



# Organic matter and sediment properties determine in-lake variability of sediment $CO_2$ and $CH_4$ production and emissions of a small and shallow lake

Leandra Stephanie Emilia Praetzel[1], Nora Plenter[2,3], Sabrina Schilling[1], Marcel Schmiedeskamp[1], Gabriele Broll[2], Klaus-Holger Knorr[1]

[1]University of Münster, Institute of Landscape Ecology, Biogeochemistry and Ecohydrology Research Group

[2]University of Osnabrück, Institute of Geography, Agroecology and Soil Research Group

[3]University of Applied Sciences Osnabrück, Faculty of Agricultural Sciences and Landscape Architecture

*Correspondence to:* Leandra S. E. Praetzel (leandra.praetzel@uni-muenster.de) or Klaus-Holger Knorr (kh.knorr@uni-muenster.de)





## Abstract

Inland waters are significant sources of $CO_2$ and $CH_4$ to the atmosphere, following recent studies this is

particularly the case for small and shallow lakes. The spatial in-lake heterogeneity of $CO_2$ and $CH_4$ production processes and their drivers in the sediment yet remain poorly studied. We thus measured potential $CO_2$ and $CH_4$ production in sediment incubations from 12 sites within the small and shallow crater lake Windsborn in Germany as well as fluxes at the water-atmosphere interface at four sites. Production rates were highly variable and ranged from 7.2 and 38.5 µmol $CO_2$ $gC^{-1}$ $d^{-1}$ and from 5.4 to

33.5 µmol $CH_4$ $gC^{-1}$ $d^{-1}$. Fluxes lay between 4.5 and 26.9 mmol $CO_2$ $m^{-2}$ $d^{-1}$ and between 0 and 9.8 mmol $CH_4$ $m^{-2}$ $d^{-1}$. Both $CO_2$ and $CH_4$ production rates and $CH_4$ fluxes were significantly negative ($p<0.05$, $rho<-0.6$) correlated with the prevalence of recalcitrant organic matter compounds in the sediment as identified by FTIR spectroscopy. The C/N ratio was significantly ($p<0.01$, $rho=-0.88$) correlated with $CH_4$ fluxes, but neither with production rates nor $CO_2$ fluxes. Inorganic (nitrate, sulfate, ferric iron) and

organic (humic acids) electron acceptors together could explain differences in $CH_4$ production rates ($R^2=0.22$) whereas we did not find clear relationships between organic matter quality, methanogenic pathways (acetoclastic vs. hydrogenotrophic) and electron accepting capacity of the organic matter. Grain size distribution could sufficiently ($p<0.05$, $rho=\pm0.65$) explain differences in $CH_4$ fluxes. Surprisingly, sediment gas storage, potential production rates and water-atmosphere fluxes were

decoupled from each other and did not show any correlations. Our results show that there exists a significant spatial variability of sediment gas production even within small lakes which can be explained by the origin and pre-processing, and therefore the degradability of the organic matter. We highlight that measuring production rates is not a suitable way to replace in-situ flux measurements as it neglects physical sediment properties and production and oxidation processes in the water column.



## 1 Introduction

Inland waters play an important role in the global carbon cycle and contribute significantly to the natural emission of the greenhouse gases carbon dioxide ($CO_2$) and methane ($CH_4$) (Cole et al., 2007; Battin et al., 2009; Bastviken et al., 2011; Raymond et al., 2013; Regnier et al., 2013). Lakes and reservoirs are
estimated to emit in total $0.32 – 0.39$ Pg C of $CO_2$ and $0.58$ Pg C ($CO_2$ -eq.) $CH_4$ year$^{-1}$ (Cole et al., 2007; Bastviken et al., 2011; Raymond et al., 2013). Especially small lakes ($< 0.1$ km²) have been underestimated in the past in regard to their spatial expansion and therefore their contribution to global greenhouse gas emissions (Downing, 2010). Although small lakes account for approx. 1/3 of the total lake area and cover less than 1% of the global land surface, they contribute 35% of $CO_2$ and 72% of
$CH_4$ emissions from lakes worldwide (Downing et al., 2006; Holgerson and Raymond, 2016). Even if the highest number of lakes can be found in boreal zones, the largest areas occur in lower latitudes around 50° (Verpoorter et al., 2014). Due to their shallowness, shorter water residence times and a smaller perimeter to volume ratio, metabolic processes and carbon (C) turnover in small lakes is much faster compared to larger lakes and they can therefore be expected to be more vulnerable to
environmental and climatic change than the latter (Wetzel, 1992; Downing, 2010).

Recently, many studies have shown that both $CO_2$ and $CH_4$ fluxes are highly variable both on a spatial and temporal scale, but the majority of these measurements has been taken out on larger lakes and is concentrated in boreal regions (Schilder et al., 2013; Wik et al., 2013; Bastviken et al., 2015; Natchimuthu et al., 2016; Natchimuthu et al., 2017; Spafford and Risk, 2018). Beyond that, studies on
gas production processes in the sediments, and studies linking sediment gas production to emissions both remain scarce. Nevertheless, anoxic sediments play a crucial role in-lake C cycling as they are the main producer of $CO_2$ and $CH_4$ which is subsequently released through the water column to the atmosphere. To understand the spatial patterns of $CO_2$ and $CH_4$ emissions, it is therefore crucial to understand $CO_2$ and $CH_4$ production processes in the sediment as well as their major controls.

The degradability of OM, and therefore the amount of produced $CO_2$ and $CH_4$, is mainly depending on its components, the microbial biomass and enzyme activities (Updegraff et al., 1995; McLatchey and Reddy, 1998; Fenchel et al., 2012). While C/N ratios can be used to determine the origin and the degradation state of OM, Fourier-transformed infrared (FTIR) spectroscopy can provide qualitative information about OM components and therefore its grade of decomposition (Meyers, 1994; Broder et
al., 2012; Biester et al., 2014; Li et al., 2016). Artz et al. (2008) compiled a range of functional moieties that is used to characterize OM in peat soils but is also applicable to OM in general. While polysaccharides and proteins are preferentially degraded by microorganisms, cellulose or aromatic compounds (e.g. lignin) are due to their molecular structure more recalcitrant and therefore accumulate in the anoxic sediment (Fenchel et al., 2012; Tfaily et al., 2014). The prevalence of different OM
compounds leads to specific FTIR spectra, as functional moieties have wavelength-specific absorption maxima (Niemeyer et al., 1992; Cocozza et al., 2003).



In anoxic sediments, $CO_2$ and $CH_4$ are produced during breakdown of OM over a cascade of microbially induced processes. After the fermentation of complex organic polymers which mainly produces hydrogen ($H_2$) and OM of low molecular weight (e.g. acetate), the latter are being further oxidized to

$CO_2$ together with a set of electron acceptors (nitrate, sulfate, ferric iron or humic substances) before $CH_4$ production initiates as the final step of terminal electron accepting processes after depletion of all other, thermodynamically more favorable electron acceptors. $CH_4$ in turn is mainly produced via two different pathways, the acetoclastic and the hydrogenotrophic pathway with either acetate (R1) or $H_2$ as substrate and $CO_2$ as electron acceptor (R2) (Conrad, 1999; Whiticar, 1999).

$CH_3COOH \rightarrow CO_2 + CH_4$ $\quad\quad\quad\quad$ $\Delta G_r^0$ (25°C) = -50.99 kJ mol$^{-1}$ $\quad\quad\quad\quad$ R1

$CO_2 + 4 H_2 \rightarrow 2 H_2O + CH_4$ $\quad\quad\quad$ $\Delta G_r^0$ (25°C) = -193.03 kJ mol$^{-1}$ $\quad\quad\quad$ R2

Several studies have shown that in anoxic wetland and marine sediments rich in labile organic compounds the acetoclastic pathway, which is energetically more favorable for microorganisms, dominates, whereas with increasing recalcitrance, the hydrogenotrophic pathway becomes more

important as acetate as direct precursor of $CH_4$ gets depleted (Schoell, 1988; Hornibrook et al., 1997; Miyajima et al., 1997). Lojen et al. (1999) suggested that seasonal changes in the dominant methane pathway in a lake sediment are attributed to OM quality.

Whereas the potential of inorganic electron acceptors to suppress methanogenic activity is well studied (Yao et al., 1999; Fenchel et al., 2012), information on the role of humic substances as organic electron

acceptors remain scarce. Klüpfel et al. (2014) revealed the potential of humic substances to be reduced and re-oxidized at oxic-anoxic interfaces in peatlands, sediments or soils underlying water table fluctuations and it has recently been shown that availability of EAC in OM controls $CO_2$ and $CH_4$ production in peat soils poor in inorganic EAs (Gao et al., 2019). As sediments from the lake under study here are also rich in OM, we wanted to verify if electron accepting (EAC) and donating capacities

(EDC) of humic substances also play a role in explaining spatial variabilities of $CO_2$ and $CH_4$ production in lake sediments that are not subjected to water table fluctuations but might to a small extent in the upper parts of the sediments be influenced by oxygen from the water column due to the in our case prevalent perennial circulation (Lau et al., 2016).

The amount of $CO_2$ and $CH_4$ produced is therefore either depending on the availability and degradability

of the OM itself, the presence of EAs and the concentration of $H_2$ and acetate as substrates for methanogenesis (Segers, 1998; Conrad, 1999; Megonigal et al., 2003; Blodau, 2011; Fenchel et al., 2012).

The spatial distribution of OM and sediment properties within lakes has been found to be highly variable in terms of their origin (terrestrial vs. aquatic), degradability, elemental geochemistry and grain size

(Muri and Wakeham, 2006; Ostrovsky and Tęgowski, 2010; Tolu et al., 2017). It has also been shown that sediment grain size is an important factor for the evolution of CH4 bubbles in sediments (Ostrovsky





and Tęgowski, 2010; Liu et al., 2018). Liu et al. (2016) for example revealed a decrease in CH4 ebullition with increased shares of sand in lake sediments. CH4 ebullition in turn is accountable for a large proportion (75%) of CH4 emissions from lakes (Bastviken et al., 2011).

So far, experimental incubations of lake sediments were only conducted with samples from one or few sites within one lake, rather comparing different lakes with each other than to study within-lake variations of $CO_2$ and $CH_4$ production and with a focus on temperature effects on production rates (Duc et al., 2010; Gudasz et al., 2010; Gudasz et al., 2015; Fuchs et al., 2016). Although a broad range of controls on $CO_2$ and $CH_4$ production has been widely investigated for anoxic peatland soils, studies on

lake sediment are rare. To our knowledge, controls such as organic matter (OM) quality, the occurrence of alternative electron acceptors (EAs), thermodynamic processes and sediment grain size have not, or only individually, been systematically surveyed.

To close this knowledge gap, we determined the magnitude and spatial variability of sediment $CO_2$ and $CH_4$ production in a small and shallow temperate lake, and connected productions patterns to OM and

sediment characteristics, thermodynamics, and water-atmosphere fluxes. To this end, we conducted slurry and intact sediment mesocosm incubations with sediment from crater lake Windsborn in Germany. This site was chosen as a model system having a high sediment OM content (~30%), a very small catchment area and no surficial in- or outflows in order to keep influences from the surrounding area as small as possible.

We hypothesize that (I) there exists a noticeable spatial variability of $CO_2$ and $CH_4$ production in the sediment, (II) the variability of production rates is reflected in the flux patterns, and that (III) production rates, methanogenic pathways and flux patterns can be explained by OM degradability, the occurrence of organic and inorganic EAs, and grain size distribution or a combination of these factors.



## 2 Materials and Methods

### 2.1 Study site

The studied Lake Windsborn is a polymictic, small and shallow crater lake in the Volcanic Eifel, Rhineland Palatinate, south-west Germany and is part of the "Mosenberg" volcano group (see Fig. 1) (LfU, 2013). The climate is temperate with mean annual temperature of 8.3°C and 931 mm precipitation (multi-annual mean 1981-2010; DWD, 2019). Lake Windsborn is the only genuine crater lake north of the Alps. It emerged approx. 29,000 years ago after a volcanic eruption when the top of the volcano was blast away and the newly formed crater was subsequently filled with water (LfU, 2013). The present lake arose around 1850, after drainage of the lake and the partial removal of the peat from the lake bottom (Kappes and Sinsch, 2005). The lake is part of a conservation area that was established in 1927 (Kappes and Sinsch, 2005). From 1950 until the 1990s, the lake was used a fishing ground, and was therefore stocked up with fish and limed (LfU, 2013). The lake is nearly circular and surrounded by a 20 to 30 meter high rampart which consists of an alternation of red-brown ashes, slag and lapilli from the eruption. Therefore, it has a very small catchment of only about 8 ha compared to the lake surface of 1.41 ha without any surficial in- and outflows and is only fed by groundwater and precipitation (LfU, 2013; Meyer, 2013). The maximum lake depth varies between 1.3 and 1.7 meters (Kappes and Sinsch, 2005). The area is underlain by devonic quartzite (Kampf, 2005).

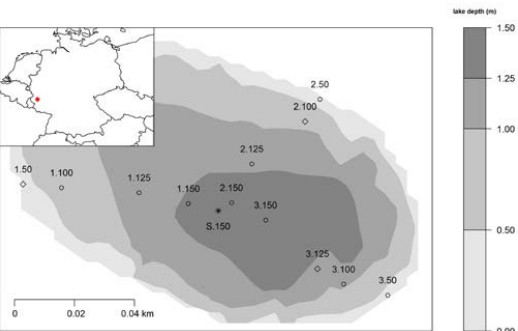

*Figure 1: Location of the study area and 13 sampling sites within Lake Windsborn. Rhombus: Sampling sites for incubations and sediment mesocosms, circles: sites for incubations only, asterisk: site for sediment mesocosm only (reference for 1.150, 2.150 and 3.150). Depths were interpolated by bivariate linear interpolation. Numbers 1, 2 and 3 refer to the number of transect from lake shore to center, numbers 50, 100, 125 and 150 indicate lake depth category (<50 cm, <100 cm, <125 cm, <150 cm).*

The lakes shoreline is vegetated with *Carex rostrata*, *Comarum palustre* and *Menynathes trifoliata*, all indicating poor nitrogen supply (Ellenberg et al., 2001). At the north-western riparian zone emerges a quaking bog of mainly *Sphagnum spec.* whose expansion will slowly lead to the silting up of the lake. Lake Windsborn was considered as humic-oligotrophic, but has transitioned to an eutrophic lake in the early 1990's now slowly recovering from human impacts and nutrient input (Kappes et al., 2000).





During our measurement campaigns in 2017 and 2018, the lake exhibited partly meso- and eutrophic features as shown in Table 1.

*Table 1: Selected lake water characteristics measured in Lake Windsborn in 2017 and 2018. pH, conductivity and O2 concentration were determined in-situ, DOC and TN were determined by catalytic oxidation, chloride was determined by ion chromatography, and other elements by inductively coupled plasma optical emission spectroscopy (ICP-OES).*

| parameter | pH | cond. | $O_2$ | Chlorophyll α | DOC | TN | $Cl^-$ |
|---|---|---|---|---|---|---|---|
| unit | | $\mu S\ cm^{-1}$ | $mg\ L^{-1}$ | $\mu g\ L^{-1}$ | | $mg\ L^{-1}$ | |
| *n* | *398* | *387* | *397* | *163* | *419* | *419* | *361* |
| average | 6.8 | 19.35 | 9.67 | 27.67 | 13.74 | 0.96 | 2.86 |
| ± SD | 0.84 | 1.72 | 0.91 | 18.66 | 2.30 | 1.89 | 3.44 |
| parameter | Ca | Fe | K | Mg | Na | P | S |
| unit | | | | $mg\ L^{-1}$ | | | |
| *n* | *379* | *378* | *379* | *379* | *379* | *329* | *379* |
| average | 1.17 | 0.11 | 0.77 | 0.72 | 4.01 | 0.06 | 0.41 |
| ± SD | 0.27 | 0.06 | 0.42 | 0.13 | 5.01 | 0.06 | 0.13 |

## 2.2 Sediment incubations

### 2.2.1 Sampling and preparation of incubations

Samples for the incubation experiment were taken at three occasions in March, April and May 2018 from in total 12 locations within the lake from three transects covering multiple water depths (<50, <100, <125, and <150 cm) (see. Fig. 1). On each sampling date, four sampling sites were chosen randomly as

it was not possible to set up the experiment with all samples at once. Measured air, water and sediment temperatures at the sampling dates were 7.7, 5.6 and 5.2 °C (March), 13.8, 15.1 and 10.1 °C (April) and 23.9, 23.6 and 13.8 °C (May). Sediment samples were taken in duplicates from a boat with a gravity corer (UWITEC, Mondsee, Austria) in 60 cm long PVC tubes and transported in an insulated box at ~5°C. Sediment cores were cut with a core cutter in the laboratory the next day in segments of 5 cm

thickness (0-5 and 5-10 cm sediment depth). Duplicate samples were homogenized and then 20 g of each sediment was filled into 120 mL crimp vials added with 20 mL lake-water, and closed with a butyl-rubber stopper and aluminum crimp cap. Samples were flushed with nitrogen for 30 minutes in order to remove remaining oxygen from water and headspace, pre-incubated for one week to have them fully anoxic and then again flushed with nitrogen prior to the actual incubation. Incubations were set up in

triplicates and stored at 25°C (corresponding to maximum measured in-situ sediment temperatures in summer 2018) in the dark. At each run, one set of parallel samples was incubated at 10°C in order to determine a $Q_{10}$-value for $CO_2$ and $CH_4$ production rates.



The remaining sample material was freeze-dried (Alpha 1-4 LPplus, Christ, Osterode, Germany),
ground with a ball mill (Mixer Mill MM 400, Retsch, Haan, Germany) and used for solid phase analyses
as outlined below.

### 2.2.2 Sediment organic matter quality

Freeze-dried and ground sediment samples were analyzed for total carbon (C), nitrogen (N) and sulfur
(S) concentrations and stable isotopes using isotope-ratio mass spectrometry (IRMS; Eurovector
EA3000 coupled with Nu Instruments Nu Horizon, Hekatech, Wegberg, Germany) and for organic
matter (OM) components using FTIR-Spectroscopy (Cary 670 FTIR Spectrometer, Agilent, Santa Clara,
USA).

For IRMS, 5 mg of sample were weighed out into a tin cup together with 4 mg of Vanadium-Pentoxide
($V_2O_5$). The combustion and reduction furnace were set to 1000 and 650 °C, respectively, the resultant
gaseous compounds were quantified by IRMS. Results are provided in % by mass for C, N and S
contents and in ‰ vs. VPDB/AIR/VCDT for C, N and S isotopic signatures. For isotope analyses,
appropriate certified reference materials were used: IAEA 600 ($\delta^{13}C$ = -27.771 ‰; $\delta^{15}N$ = 1.0 ‰) and
S-1 ($\delta^{34}S$ = -0.30 ‰) for calibration, and BBOT (2.5-Bis-(5-tert.-butyl-2-benzo-oxazol-2-yl)thiophen;
Hekatech, Wegberg, Germany), birch leaf, wheat flour and sorghum flour standards (IVA
Analysetechnik e. K., Meerbusch, Germany) as working standards covering a range of -27.5 to -13.68
‰ for $^{13}C$, -0.6 to 2.12 ‰ for $^{15}N$, and -9.3 to -1.42 ‰ for $^{34}S$.

Functional groups of OM compounds were identified by FTIR spectroscopy. Therefore, 2 mg of freeze-
dried sample were ground together with 200 mg KBr (potassium bromide) in a mortar, pressed to 13
mm pellets and analyzed. Each sample was scanned from 599 to 4000 $cm^{-1}$ with a resolution of 2 $cm^{-1}$
and baseline corrected. Distinct peaks at specific wavelengths were assigned to functional groups
according to Artz et al. (2008) and normalized to the peak intensity at 1031 - 1035 $cm^{-1}$ (indicative of
polysaccharides) in order to obtain inter-comparable peak-ratios of functional moieties in all samples as
FTIR spectra only provide information about relative abundance of certain functional moieties.

### 2.2.3 $CO_2$ and $CH_4$ production rates from incubations

Potential $CO_2$ and $CH_4$ productions rates were determined by measuring the increase in concentration
of $CO_2$ and $CH_4$ in the incubation vials over time. Concentrations were obtained from analysis of the
headspace at the beginning of the experiment (t0), and after 1, 3, 8, 11, 14 and 18 days (t1-t6). Samples
were taken from the vial with a 10 mL PP syringe equipped with a 3-way-stopcock and 0.6 mm needle.
Before each sampling, the pressure inside the vial was determined with a pressure sensor (GMH 3110,
Greisinger, Regenstauf, Germany), and the syringe was three-times flushed with nitrogen, then 2 mL of
nitrogen were left in the syringe before stabbing through the stopper, the nitrogen was added to the
headspace, mixed, and subsequently 2 mL of sample were taken from the vial so that the volume inside
the vial remained constant. The gas samples were analyzed for $CO_2$ and $CH_4$ concentrations with a gas
chromatograph (8610 GC-TCD/FID, SRI Instruments, Torrance, USA) equipped with a Flame



Ionisation Detector (FID) and Methanizer to simultaneously measure $CO_2$ and $CH_4$. The gas

chromatograph was calibrated with standard gas mixtures of known concentrations ($CO_2$: 385, 5,000 and 50,000 ppmV; $CH_4$: 5, 1,000 and 50,000 ppmV) before every sampling day.

First, measured concentrations in ppmV were pressure corrected and converted using the ideal gas law:

$$n = (p * V) / (R * T) \tag{1}$$

where n is the amount of substance in mol, p is the gas partial pressure in atm, V is the headspace volume

in L, R is the ideal gas constant (0.082 L atm $mol^{-1}$ $K^{-1}$) and T is the laboratory temperature in K.

Total $CO_2$ and $CH_4$ concentrations in gas and water phase in the incubation vials were calculated from headspace concentrations with Henry's Law:

$$c = K_h * p \tag{2}$$

where c is the concentration in the water phase in mol $L^{-1}$, $K_h$ is the temperature-dependent Henry-

constant ($CO_2$, 25°C = 0.0339 mol $L^{-1}$ atm $^{-1}$, $CH_4$, 25°C = 0.00129 mol $L^{-1}$ atm $^{-1}$ (Sander, 2015)) and p is the gas partial pressure in atm.

Moreover, $CO_2$ concentrations were pH-corrected in order to obtain pH-independent values for total $CO_2$ concentrations using Henderson-Hasselbalch-equation and equilibrium constants according to Stumm and Morgan (1996):

$$n_{\Sigma CO2} = n_{water} * 10^{pH-6.4} + (n_{water} * 10^{pH-6.4}) * 10^{pH-10.25} \tag{3}$$

where $n_{\Sigma CO2}$ is the pH-corrected $CO_2$ amount in mol, and $n_{water}$ is the calculated amount of $CO_2$ in the water phase in mol.

Finally, production rates were calculated as linear regression ($R^2$ > 0.8 for $CO_2$ and > 0.9 for $CH_4$) from concentration change over time.

To evaluate the effect of temperature on $CO_2$ and $CH_4$ production, we calculated $Q_{10}$-values, describing the relative increase of production rates with an increase in temperature of 10 Kelvin (Fenchel et al., 2012).

$$Q_{10} = (R2/R1)^{[10/(T2-T1)]} \tag{4}$$

where R2 is the production rate at T2 (25°C), and R1 is the production rate at T1 (10°C).

### 2.2.4 Thermodynamics and methanogenic pathways

In order to calculate the thermodynamic energy yield for hydrogenotrophic and acetoclastic methanogenesis we measured hydrogen ($H_2$) concentrations at the beginning (t0), after 8 days (t3) and at the end (t6) of the experiment and acetate ($H_3COO^-$) concentrations at the beginning and at the end of



the incubation. The thermodynamic energy yield, expressed as Gibb's free energy was calculated using

the Nernst equation as described in Beer and Blodau (2007):

$$\Delta G_r = \Delta G_r{}^0 + R * T \ln (\Pi_i(products)^{vi} / \Pi_i(educts)^{vi}) \tag{5}$$

By calculating the Gibb's free energy ($\Delta G_r$), it is possible to evaluate whether these processes are feasible under given conditions. Therefore a theoretical threshold of -20 to -25 kJ mol$^{-1}$ for has to be exceeded (Schink, 1997; Conrad, 1999; Blodau, 2011).

For $H_2$ concentration measurements, 2 mL of sample were taken from the incubation headspace with a syringe and needle and replaced with the same amount of N2. Samples were analyzed with a Reduction Gas Detector (RGD) Hydrogen and Carbon Monoxide Analyzer (ta3000R Gas Analyzer, Ametek, Pittsburgh, USA) that was calibrated with gas standards of 5, 25, and 50 ppmV $H_2$. Measured $H_2$ concentrations were corrected for pressure and converted into dissolved concentrations using Henry's

Law ($K_h(H_2, 25°C) = 0.00078$ mol L$^{-1}$ atm$^{-1}$ (Sander, 2015)) analogous to $CO_2$ and $CH_4$.

Acetate concentrations were determined by ion chromatography IC with chemical suppression (883 Basic IC plus, Metrohm, Filderstadt, Germany; A-supp 5 column, Metrohm, Filderstadt, Germany). Aqueous samples were filtered with 0.45 µm Nylon + Glass Micro Fibre syringe filters (Simplepure, BGB Analytik, Rheinfelden, Germany) and kept frozen at -21°C until analysis.

2.2.5 Alternative Electron Acceptors

To quantify alternative Electron Acceptors (EAs) that could support anaerobic respiration and potentially suppress methanogenesis in anoxic incubations, we analyzed for nitrate ($NO_3^-$), sulfate ($SO_4^{2-}$), iron (III) ($Fe^{3+}$), and electron accepting and donating capacity of the organic matter (EAC/EDC$_{OM}$) at the beginning (t0) and the end (t6) of the incubation.

As the analysis of EAC$_{OM}$ and EDC$_{OM}$ is so far only possible on finely ground materials, providing potential capacities rather than true in-situ capacities, a second set of incubations for a sediment depth of 0-5 cm with samples from 10 sites (except 3.50 and 3.100) was set up with 0.4 g of the freeze-dried sediment material and 100 mL of Millipore water. Incubations were set up in six replicates, flushed with N2, pre-incubated and stored analogous to the first set of incubations. Samples were measured at the

beginning and at the end of the experiment, whereas at every sampling occasion, three of the replicates had to be sacrificed (destructive sampling). To this end, samples were transferred into a glovebox ($O_2 <$ 1 ppm, Innovative Technology, Amesbury, USA) prior to analysis to avoid alteration of the samples' redox state.

EAC$_{OM}$ and EDC$_{OM}$ were measured chronoamperometrically (CHI1000C, CH Instruments, Austin,

USA) by mediated electrochemical reduction (MER) and oxidation (MEO) (Aeschbacher et al., 2010; Klüpfel et al., 2014). The cell consisted of a cylindrical glassy carbon working electrode, a Platinum wire counter electrode in a glass-ceramic frit, and an Ag/AgCl reference electrode. Cells were filled



with 10 mL of 0.01 M/0.1 M MOPS/KCl-Buffer to stabilize the pH at 7 and continuously stirred during measurement. To facilitate electron transfer, organic mediators were added to the buffer; 180 µL DQ (diaquat-dibromide monohydrate, Sigma-Aldrich) for MER, and 180 µL ABTS (2,2'-azino-bis(3-ethylbenzthiazoline-sulfonic acid), Sigma-Aldrich) for MEO at a potential of $E_h$ = -0.69 V and $E_h$ = +0.41 V respectively (reported vs. the standard hydrogen electrode but experimentally measured vs. the Ag/AgCl reference electrode). To determine the electron transfer, 100 µL of suspended samples were added to the buffer solution (Lau et al., 2015), which resulted in an increase of the current recorded as a peak in the analysis software. After approx. 30 minutes, when the baseline was reached again, the next sample was added to the cells. Samples were measured in duplicates. The electron transfer was calculated with the Nernst-equation and normalized to the C content in the samples (Lau et al., 2015).

$$EAC_{OM} / EDC_{OM} = \text{peak area} / (V * F * C) \tag{6}$$

with $EAC_{OM}$ /$EDC_{OM}$ in µmol e⁻ gC⁻¹, peak area in µA sec, V = sample volume in µL, F = Faraday constant 96,485 A sec / mol e⁻, and C = carbon content in mg L⁻¹.

$EAC_{OM}$ and $EDC_{OM}$ had to be corrected for $Fe^{3+}$, or iron (II) ($Fe^{2+}$) and sulfide ($S^{2-}$) concentrations respectively as with the applied potential also $Fe^{3+}$ would be reduced and $Fe^{2+}$ and $S^{2-}$ oxidized (Lau et al., 2015; Agethen et al., 2018).

$Fe^{2+}$, $Fe^{3+}$ and $S^{2-}$ were determined colorimetrically (Gilboa-Garber, 1971; Tamura et al., 1974) with a spectrophotometer (Cary 100 UV-Vis, Agilent, Santa Clara, USA).

$NO_3^-$ and $SO_4^{2-}$ concentrations were determined with IC, as described above. Therefore, samples were filtered with a 0.45 µm syringe filter, filled in micro-centrifuge tubes, retrieved from the glovebox and frozen at -21°C until analysis.

Total electron accepting capacity ($EAC_{tot}$) was calculated as the sum of $EAC_{OM}$ and $EAC_{inorg}$ (EAC from nitrate, sulfate, iron (III)) considering the respective amounts of electrons transferred during main pathways of dissimilatory reduction (Konhauser, 2009):

$$EAC_{tot} = EAC_{OM} + NO_3^- * 5e^- + SO_4^{2-} * 8e^- + Fe^{3+} * 1e^- \tag{7}$$

In theory, $EAC_{OM}$ and $EDC_{OM}$ correlate with each other: if $EAC_{OM}$ decreases, $EDC_{OM}$ increases equivalently as quinones are reduced to hydroquinones. But in practice, values of $EDC_{OM}$ are potentially biased as MEO does not only capture $EDC_{OM}$, but may also irreversibly oxidize phenolic moieties, which are sensitive to slightest changes in pH and potentials (Aeschbacher et al., 2011; Walpen et al., 2016; Walpen et al., 2018). The discussion will therefore focus on $EAC_{OM}$ data.

### 2.3 Sediment mesocosms: $CO_2$ and $CH_4$ fluxes and sediment gas stock change

To obtain ex-situ $CO_2$ and $CH_4$ gas fluxes and estimate changes in sediment $CO_2$ and $CH_4$ stocks, intact sediment cores (PVC tubes, 60 cm length, 5.8 cm diameter) were taken in triplicates from four sites out of the twelve above in November 2017 (1.50, 2.100, 3.125, S.150; see Fig. 1). S.150 was chosen as one



site representing the sites 1.150, 2.150 and 3.150 from the same lake depth category. Sediment cores were transported cooled and deployed in a climate chamber (CLF Plant Master, CLF Plant Climatics GmbH, Wertingen, Germany) at constant conditions (temperature 20°C, humidity 60 %). Cores were

taken to ensure that each tube contained a sediment layer of on average 35 cm thickness, covered with a lake water column of 20 cm; in the lab we created a headspace of approx. 150 mL. The cores were equipped with eight sampling ports; one in the headspace, one in the water phase and six in the sediment. Of the latter, three were used to sample dissolved gases in the sediment, and three for sediment pore water extraction. The gas samplers consisted of a gas permeable silicon tubes of 5 cm length and 0.8 cm

diameter equipped with a 3-way-stopcock, modified after Kammann et al. (2001). This technique allows for sampling of dissolved gases in the sediment by diffusive equilibration through the silicon membrane. For pore water sampling, a vacuum was applied with a syringe to suction samplers (Rhizon, Eijkelkamp Agrisearch, Giesbeek, Netherlands) of 5 cm length, 0.25 cm diameter, and about 0.1 µm pore size (Seeberg-Elverfeldt et al., 2005). Gas and pore water samplers were deployed in average depths of 5.0

$\pm$ 2.8, 15.3 $\pm$ 2.9, and 23.6 $\pm$ 2.1 cm below sediment surface.

All measurements were conducted after 50 days of incubation of the cores in the climate chamber when the system had reached a steady state as indicated by quasi-constant $CO_2$ and $CH_4$ concentrations in the sediment.

For the determination of $CO_2$ fluxes, the cores were closed gas-tight with a stopper and connected to a

laser-based, portable greenhouse-gas analyzer (Los Gatos Research, San Jose, USA) which allowed to measure real-time increase of $CO_2$, $CH_4$ and $H_2O$ concentrations in the headspace of the cores with a resolution of 1 Hz. As the headspace was too small for the instrument's flow rate, a gas bag with a volume of 150 mL was interposed between the headspace and the analyzer. The headspace was closed for 10 minutes, and the diffusive $CO_2$ flux was calculated by linear regression ($R^2 > 0.8$) of the increase

in concentration over time and the ideal gas law, corrected for air pressure and temperature and related to the water surface area:

$$F = \Delta c / \Delta t * (p * V) / (A * R * T) \qquad (8)$$

where F is the $CO_2$ flux in µmol m$^{-2}$ d$^{-1}$, $\Delta c / \Delta t$ is the slope of the linear regression in ppm d$^{-1}$, p is the air pressure in atm, V is the sum of headspace and gas bag volume in m³, A is the water surface area in

m², R is the ideal gas constant 8.2*10$^{-5}$ m³ atm mol$^{-1}$ K$^{-1}$, and T is the temperature in K.

$CH_4$ fluxes were determined by closing the cores with the stopper for 24 hours, and taking a gas sample right after closing, and again after 24 hours with a syringe from the headspace. $CH_4$ fluxes were calculated according to Bastviken et al. (2004):

$$F(CH_4) = k * (C_w - C_{fc}) \qquad (9)$$



where F is the $CH_4$ flux in mmol $m^{-2}$ $d^{-1}$, k is the piston velocity in m $d^{-1}$, $C_w$ is the measured $CH_4$ concentration in the water phase in mmol $m^{-3}$ and $C_{fc}$ is the $CH_4$ equilibrium concentration in the headspace at the given $CH_4$ water concentration.

The piston velocity k was determined as:

$$k = (-\ln((c_{sat} - c_{end})/(c_{sat} - c_{start}))/ \Delta t * V) / (A * K_h * R * T) \tag{10}$$

where $c_{sat}$ is the saturation concentration in the chamber headspace at the measured $CH_4$ water concentration, $c_{end}$ is the measured $CH_4$ concentration in the chamber headspace at the end of the flux measurement, $c_{start}$ is the measured $CH_4$ concentration in the chamber headspace at the beginning of the flux measurement (all in µatm).

The flux was corrected for the non-linear increase of $CH_4$ concentration in the headspace over time due
to saturation and divided into diffusive and ebullitive proportions based on the piston velocity (k < 2 = diffusion, k > 2 = ebullition).

$CH_4$ and $CO_2$ concentrations in the sediment were obtained from gas-permeable silicon tubes, determined by gas chromatography as described above (2.2.3) and calculated by Henry's Law using temperature corrected Henry's constants (see formula 4). Measured $CO_2$ concentrations were corrected
for pH (formula 5).

The storage change of $CO_2$ and $CH_4$ in the sediment was calculated for each depth segment between two sampling ports as the difference between $CH_4$ concentrations obtained from silicon gas samples at the beginning and at the end of gas flux measurements:

$$\Delta CO_2/CH_4 = ((c(CO_2/CH_4)_{end} * V_{seg}) - (c(CO_2/CH_4)_{start} * V_{seg})) / \Delta t \tag{11}$$

where $\Delta$ $CO_2/CH_4$ is the storage change in mmol $d^{-1}$, $c(CO_2/CH_4)_{end/start}$ is the $CO_2$ / $CH_4$ sediment pore gas concentration at the end/beginning of the flux measurement in mmol $m^{-3}$ and $V_{seg}$ is the volume of the sediment core segment between two samplers in $m^3$.

After completion of flux measurements, sediment mesocosms were eventually cut into 10 cm slices, freeze-dried and ground for solid phase analyses as described above.

## 2.4 Other chemical and physical parameters in water and sediment

Total phosphorus (P), sulphur (S), manganese (Mn) and iron (Fe) in the sediment were determined by wavelength dispersive X-ray fluorescence (WD-XRF; ZSX Primus II, Rigaku, Tokyo, Japan). To this end, 500 mg of freeze-dried and ground sample were pressed to pellets together with 50 mg of wax (Hoechst Wax C, Merck, Darmstadt, Germany) as pelleting agent. Calibration of the instrument was
done using a set of 22 certified reference materials.

Dissolved organic carbon (DOC) and total nitrogen (TN) concentrations in watery samples and sediment pore water were determined by catalytic oxidation and subsequent NDIR detection with a total



carbon/nitrogen analyzer (TOC-L/TNM-L, Shimadzu, Kyoto, Japan). Calibration was verified at each measurement day with potassium hydrogen phthalate (5 and 25 mg L$^{-1}$) and potassium nitrate (1 and 10

mg L$^{-1}$) standard solutions.

Grain size distribution was determined after Austrian standards (OENorm B 4412; OENorm L 1050; OENorm L 1061) by the Physio-geographic Lab of the Institute of Geography and Regional Research of the University of Vienna. To this end, the organic substance was removed from the samples with hydrogen peroxide (H$_2$O$_2$) prior to analyses and mineral fine soil was divided into clay (< 2 µm), silt

(fine (2-6 µm), medium (6-20 µm), coarse (20-63 µm)) and sand (fine (63-200 µm), medium (200-630 µm), coarse (630-2000 µm)).

### 2.5 Statistics

All statistical analyses were conducted with R Studio, Version 3.5. 2 (R Core Team, 2018). Data was tested for normal distribution and homoscedasticity with Shapiro-Wilk and Levene- Test (Fox and

Weisberg, 2011) respectively. For non-normally distributed data, significant differences between groups were identified using Kruskal-Wallis and Post-hoc Dunn- Test (Dinno, 2017) for more than two groups and Mann-Whitney-Test for comparing two groups. If the condition of homoscedasticity was not fulfilled, groups were compared with Mood's Median Test. Correlations and regressions between production rates and sediment parameters were calculated by Spearman's Rank Correlation and by linear

regression models respectively. All data was tested on a 95% confidence interval and significance level of $\alpha = 0.05$.





# 3 Results

## 3.1 Sediment incubations

### 3.1.1 Sediment Organic Matter Quality

The C content in the samples was between 2.15 and 33.16% with lowest values at site 3.50 and highest at site 1.50. C/N ratios ranged from 10.97 at site 1.150 to 19.06 at site 3.100. Neither C content nor C/N ratio showed significant changes with sediment nor lake depth, but C/N ratio was significantly higher in samples taken close to the shore (50) than in samples from the lake center (150) ($p < 0.01$).

Organic matter quality as identified by FTIR-analysis was predominated by strong absorption features of polysaccharides, lignin, humic acids, phenolic and aliphatic structures, aromatic compounds, and fats, waxes and lipids (see Fig. S1). Except for lignin, which was not identified at sites 2.50, 2.100, 2.125, 3.50, and 3.100, all components were found at all sites. Ranges of peak ratios for every component can be found in Tab. 2. Overall, lowest peak ratios were present at site 3.50 and highest at site 3.125, corresponding to highest and lowest $CH_4$ production rates. All peak ratios correlated with each other. They tended to increase with sediment depth and towards the lake center, respectively, but this change was not significant.

*Table 2: C, N, P, S, Mn, and Fe contents, C/N and FTIR peak ratios of identified OM compounds and their minimum and maximum values. n = 16-22.*

|  | Min | Max | Average |
|---|---|---|---|
| **C (%)** | 2.15 | 33.16 | 26.45 |
| **N (%)** | 0.15 | 2.45 | 1.97 |
| **C/N ratio** | 10.97 | 19.06 | 13.81 |
| **$\delta^{13}$C (‰)** | -27.84 | -27.3 | -27.6 |
| **$\delta^{15}$N (‰)** | -6.27 | 1.33 | -0.93 |
| **P (%)** | 0.3 | 0.51 | 0.37 |
| **S (%)** | 0.13 | 1.22 | 0.93 |
| **Mn (%)** | 0.037 | 0.12 | 0.049 |
| **Fe (%)** | 2.13 | 7.01 | 2.95 |
| **Lignin (1220-1234 cm$^{-1}$)** | 0 | 0.437 | 0.235 |
| **Humic acids (1417-1419 cm$^{-1}$)** | 0.039 | 0.491 | 0.328 |
| **Phenols & aliphatics (1456 cm$^{-1}$)** | 0.018 | 0.480 | 0.324 |
| **Other Aromatics (1623-1646 cm$^{-1}$)** | 0.210 | 0.825 | 0.629 |
| **Fats, waxes, lipids (2850-2856 cm$^{-1}$)** | 0.178 | 0.492 | 0.349 |





### 3.1.2 $CO_2$ and $CH_4$ production rates

Rates for potential $CO_2$ production in 0-5 cm depth ranged from $10.309 \pm 0.003$ µmol gC$^{-1}$ d$^{-1}$ at site 3.125 to $38.515 \pm 2.48$ µmol gC$^{-1}$ d$^{-1}$ at site 2.100. Potential $CH_4$ production lay between $7.3 \pm 0.14$ µmol gC$^{-1}$ d$^{-1}$ at site 3.125 and $33.53 \pm 7.151$ µmol gC$^{-1}$ d$^{-1}$ at site 3.50. Production rates in 5-10 cm
depth were always lower compared to the upper sediment layer and were between $7.189 \pm 0.072$ and $14.714 \pm 0.435$ µmol gC$^{-1}$ d$^{-1}$ for $CO_2$ and between $5.361 \pm 0.259$ and $14.264 \pm 0.341$ gC$^{-1}$ d$^{-1}$ for $CH_4$. Both $CO_2$ and $CH_4$ production rates showed significant differences between sites (Kruskal-Wallis, $p < 0.001$) and were significantly (Dunn's, $p < 0.05$) higher at the shore (50+100) than in the center of the lake (125+150) (see Fig. 2 + Fig. S2).

$CO_2$ and $CH_4$ production rates decreased with time. $CO_2$ production was highest at the beginning, while $CH_4$ production had its peak after three to eight days of incubation (s. Fig. 3, Tab. 3). The $CO_2$/ $CH_4$ ratio constantly decreased during the incubation, with a maximum value of $62.13 \pm 58.44$ at the beginning of the incubation at site 1.125 and minimum values of $1.17 \pm 0.004$ at the end of the incubation at site 2.50 (s. also Tab. 3).

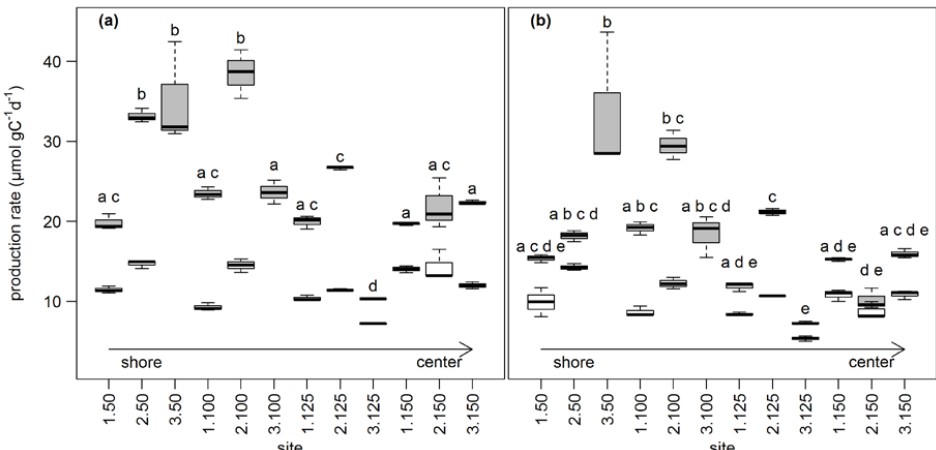


***Figure 2:*** *$CO_2$ (a) and $CH_4$ (b) production rates in 0-5 (grey) and 5-10 (white) cm sediment depth. n = 3. Production rates are calculated from linear regression of concentration increase over time. Bold lines are the median, boxes show the 25 and 75 percentile, and whiskers indicate minima and maxima within 1.5 times the interquartile range. Different letters indicate significant differences between these sites.*





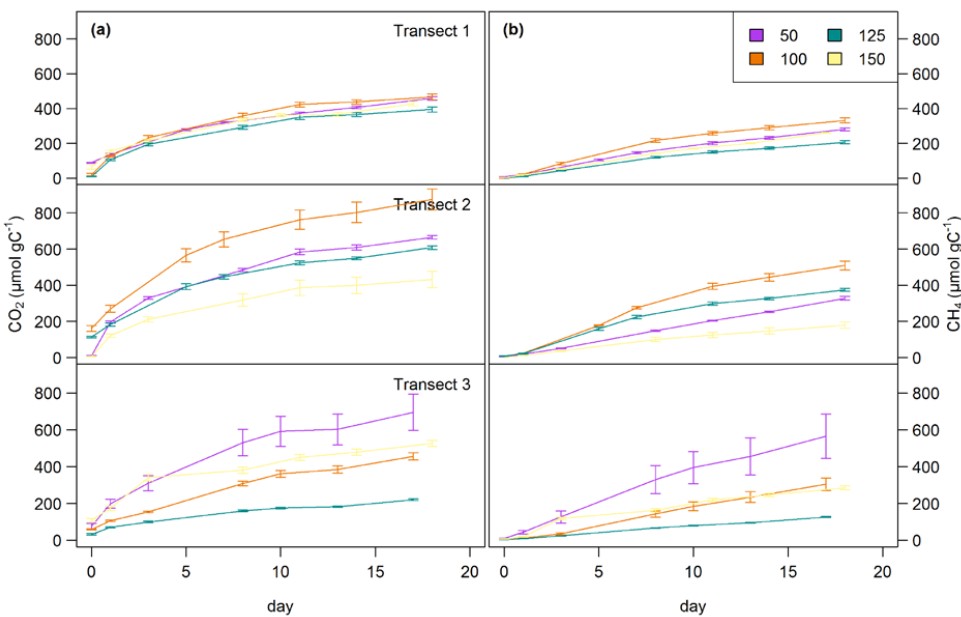


***Figure 3:*** *$CO_2$ (a) and $CH_4$ (b) production over time at all sites in 0-5 cm sediment depth. Top: transect 1, middle: transect 2, bottom: transect 3. Line are average values of triplicate measurements ± SD.*

***Table 3:*** *$CO_2$ and $CH_4$ production rates and $CO_2$/ $CH_4$ ratio over time. Rates are in μmol $gC^{-1}$ $d^{-1}$, ratio is in μmol. Values in brackets are SDs.*

| | sediment depth | t0 | t1 | t2 | t3 | t4 | t5 | t6 |
|---|---|---|---|---|---|---|---|---|
| **$CO_2$ production rate** | 0-5 cm | | 91.49 (40.29) | 44.67 (16.77) | 26.49 (9.69) | 21.14 (7.59) | 6.79 (3.59) | 13.32 (5.00) |
| | 5-10 cm | | 39.86 (14.18) | 19.63 (5.95) | 11.01 (2.69) | 11.40 (3.06) | 6.00 (2.20) | 8.91 (2.74) |
| **$CH_4$ production rate** | 0-5 cm | | 14.37 (7.44) | 22.80 (11.26) | 23.88 (11.47) | 16.74 (7.86) | 11.60 (4.69) | 13.58 (5.78) |
| | 5-10 cm | | 9.36 (2.67) | 11.68 (2.85) | 11.56 (3.13) | 9.72 (2.34) | 7.89 (2.10) | 9.32 (2.72) |
| **$CO_2$/$CH_4$ ratio** | 0-5 cm | 14.55 (6.71) | 8.70 (2.26) | 3.84 (1.32) | 2.33 (0.48) | 2.11 (0.45) | 1.89 (0.37) | 1.73 (0.30) |
| | 5-10 cm | 16.19 (24.90) | 5.61 (1.60) | 2.92 (0.88) | 1.93 (0.35) | 1.74 (0.34) | 1.57 (0.25) | 1.44 (0.20) |


$Q_{10}$-values were between $1.56 \pm 0.13$ and $2.19 \pm 0.14$ for $CO_2$ production rates and between $2.65 \pm 0.48$ and $11.37 \pm 0.96$ for $CH_4$.

Sampling dates did not have any impact on production rates as confirmed by Kruskal-Wallis-test ($CH_4$: $p = 0.173$, $CO_2$: $p = 0.755$)




Neither $CO_2$ nor $CH_4$ production rates were correlated with C content or C/N ratio, but we found significant negative correlations ($p < 0.01$, rho $< -0.6$) between all peak ratios and $CO_2$ and $CH_4$ production rates as well as with Q10-values of $CH_4$ production ($p < 0.05$, rho $= -0.82$) (see Fig. 4).

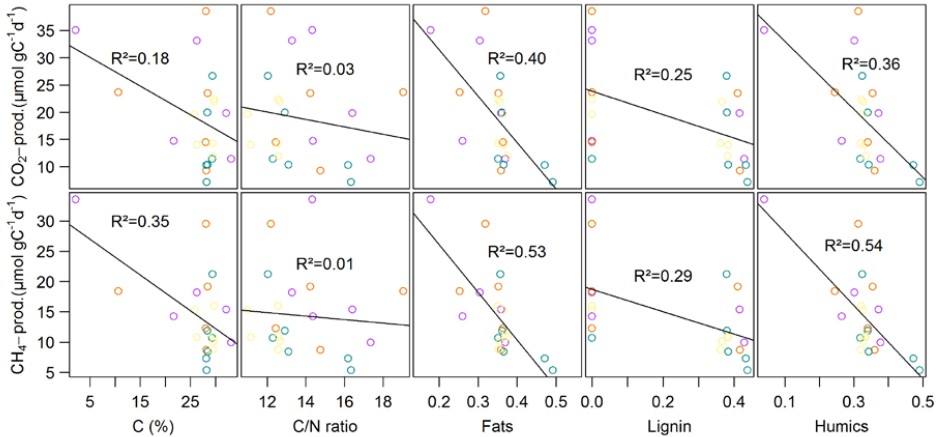

***Figure 4:*** *Correlations between $CO_2$ and $CH_4$ production rates and OM quality parameters.*


### 3.1.3 Methanogenic pathways

Both $H_2$ and acetate concentrations increased during the incubation. $H_2$ concentrations were between 0 and $1587.42 \pm 170.12$ nmol $L^{-1}$ and acetate concentrations ranged from $219.74 \pm 104.25$ to $1212.71 \pm 11.35$ µmol $L^{-1}$ (see Fig. 5). Gibb's free energy for acetoclastic methanogenesis was between $-96.32 \pm$
$0.84$ and $-66.37 \pm 0.28$ kJ $mol^{-1}$ and for hydrogenotrophic methanogenesis between $-62.41 \pm 0.85$ and $-7.41 \pm 0$ kJ $mol^{-1}$. Energy yields decreased for the acetoclastic pathway and significantly (Mood's median, t0-t6: $p < 0.05$, t3-t6: $p < 0.001$) increased for the hydrogenotrophic pathway throughout the incubation (see Fig. 5). Energy yields did not differ between lake depths except for the acetoclastic pathway at the end of the incubation (t6, $p < 0.001$), when the energy yield was highest in samples from
the center of the lake. Significant differences between all sites were found for the acetoclastic and hydrogenotrophic pathway at the beginning of the incubation (Kruskal-Wallis, $p < 0.01$) and at t6 for acetoclastic methanogenesis only (Kruskal-Wallis, $p < 0.01$). $H_2$ concentrations at the end of the experiment were significantly correlated with average $CO_2$ ($p < 0.001$, rho $= +0.51$) and $CH_4$ ($p < 0.001$, rho $= +0.45$) production rates. Further, Gibb's free energy of acetoclastic methanogenesis was positively
correlated with C/N ratio ($p < 0.05$, rho $= +0.45$) at the end of the experiment (t6).





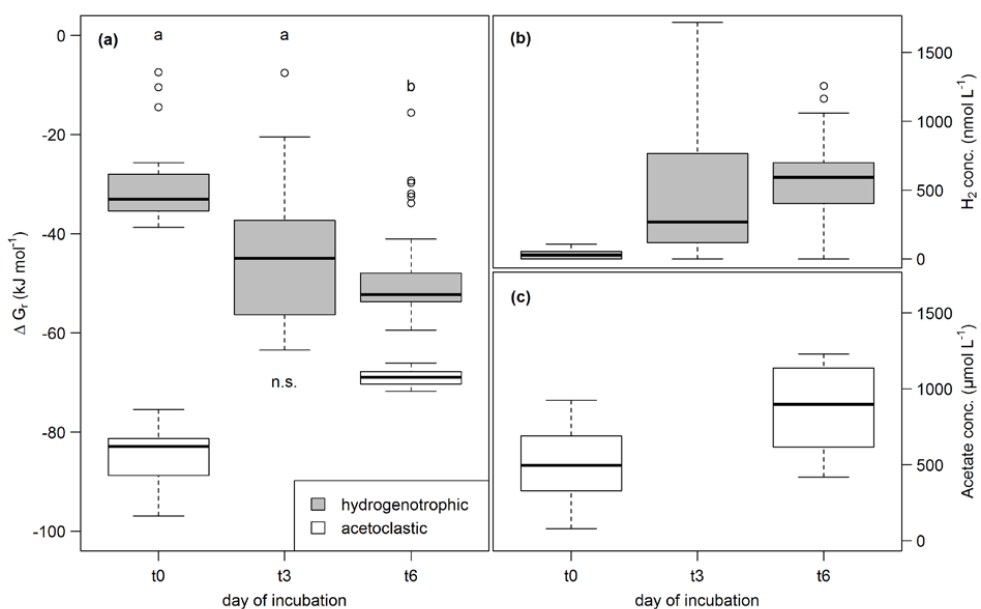

***Figure 5:*** *(a) Change of Gibb's free energy of hydrogenotrophic (grey) and acetoclastic (white) methanogenesis and of $H_2$ (b) and acetate (c) concentrations over time. More negative $\Delta G_r$ values correspond to a higher energy yield. Different letters denote significant differences between sampling dates.*


### 3.1.4 Alternative Organic and Inorganic Electron Acceptors

$EAC_{OM}$ lay between $218.69 \pm 97.15$ and $545.71 \pm 60.33$ µmol e⁻ gC⁻¹ at t0 and decreased by an average of 44.85 µmol e⁻ gC⁻¹ to $170.76 \pm 32.70$ to $460.79 \pm 51.47$ µmol e⁻ gC⁻¹ at t6 with highest values found at site 3.125 corresponding to lowest measured $CH_4$ production rates at that site. $EAC_{inorg}$ was, with

values between $19.82 \pm 10.52$ and $218.01 \pm 24.79$ µmol e⁻ gC⁻¹, lower than $EAC_{OM}$ and also decreased during the incubation by a mean of 42.40 µmol e⁻ gC⁻¹. $EAC_{tot}$ ranged from $226.95 \pm 83.87$ µmol e⁻ gC⁻¹ at site 3.150 to $614.58 \pm 80.72$ µmol e⁻ gC⁻¹ at site 3.125 and significantly (t-test, $p < 0.001$) decreased by averagely 87.25 µmol e⁻ gC⁻¹ from t0 to t6. $EDC_{OM}$ was between $149.54 \pm 26.72$ and $462.69 \pm 18.57$ µmol e⁻ gC⁻¹ at the beginning and between $152.94 \pm 53.78$ and $370.65 \pm 196.22$ µmol e⁻

gC⁻¹ at the end of the incubation with a slight increase by on average 31.38 µmol e⁻ gC⁻¹. Lowest $EDC_{OM}$ at both t0 and t6 was found at site 2.100, corresponding to highest $CO_2$ production rates there (see Fig. 6).

We further found significant differences for $EAC_{OM}$ (ANOVA, $p < 0.001$) and $EAC_{tot}$ (ANOVA, $p < 0.01$) between all sites with highest average values at site 3.125 and lowest at site 3.150 (see Fig. S3).




Average $CO_2$ and $CH_4$ production rates were significantly ($p < 0.05$, rho = 0.7) negative correlated with initial $EDC_{OM}$, whereas we did not find any significant correlation with EAC, EEC nor EAC/EDC ratio, although low $CH_4$ production rates were associated with high EAC.

$CO_2$, $CH_4$ and acetate concentration were significantly ($p < 0.05$, $< 0.05$, $< 0.001$) negative correlated with $EAC_{tot}$. Gibb's free energy of hydrogenotrophic methanogenesis was significantly ($p < 0.05$)
positive and acetate concentration significantly ($p < 0.01$) negative correlated to $EAC_{OM}$.

We did not find any significant correlations between EAC or EDC and OM quality parameters except for the FTIR peak ratio indicative of fats and lipids and $EDC_{OM}$ ($p < 0.05$, rho = -0.76).

Of the inorganic EAs, only total sulfur content in the sediment was significantly negative correlated with $CH_4$ production rates ($p < 0.05$).

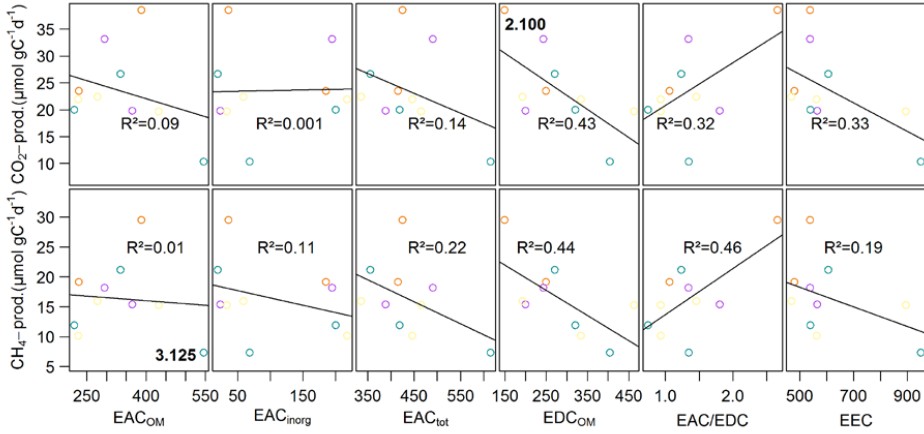


**Figure 6:** *Linear regression of $CH_4$ and $CO_2$ production rates with initial electron accepting and donating capacities. Points are mean values of triplicate measurements at each site.*

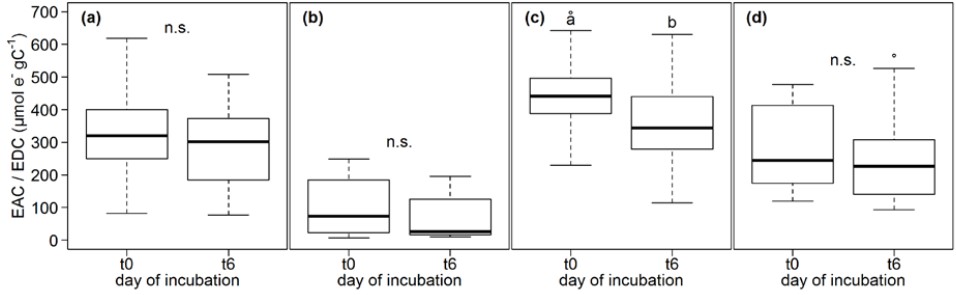

**Figure 7:** *$EAC_{OM}$ (a), $EAC_{inorg}$ (b), $EAC_{tot}$ (c) and $EDC_{OM}$ (d) at the beginning and the end of the incubation at all*
*sites. n = 60-88. Different letters denote significant differences.*

## 3.2 Sediment mesocosms: Fluxes, sediment storage change and grain size

$CO_2$ and $CH_4$ fluxes measured from sediment mesocosms ranged from 4.46 to 26.9 and 0 to 9.8 mmol m$^{-2}$ d$^{-1}$ respectively. $CH_4$ ebullition played a major role in cores from sites 2.100 and S.150 and accounted for up to 100% of the fluxes there. $CO_2$ fluxes at site 3.125 were significantly (t-test, $p < 0.05$) lower than at the other sites while total $CH_4$ fluxes at site 2.100 were significantly (Kruskal-Wallis, $p < 0.01$) higher compared to all other sites. Fluxes were within the same range like potential production rates in incubations but did only partly follow the pattern observed in incubations (see Fig. 8).

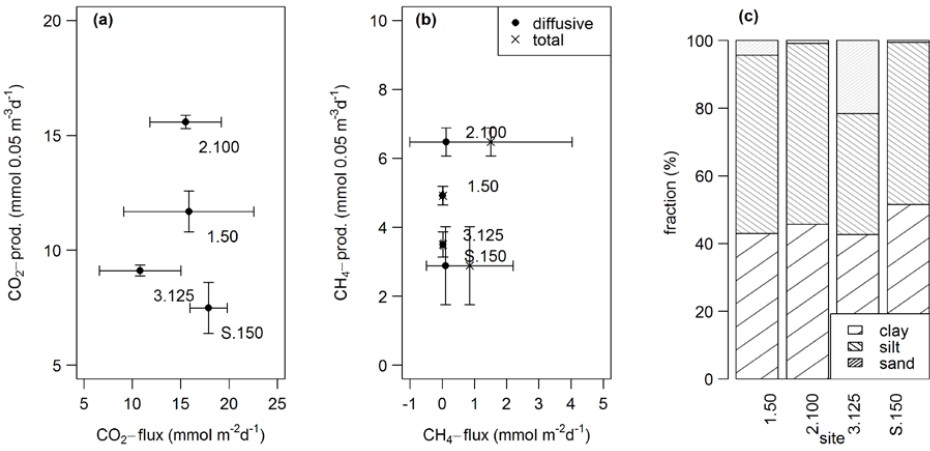

***Figure 8:*** *Comparison of fluxes obtained from sediment mesocosms and potential production rates from the incubations (in 0-5 cm sediment depth) for $CO_2$ (a) and diffusive and total $CH_4$ (b). Points are averages (n(incubations) = 3-9, n($CO_2$ fluxes) = 9-14, n($CH_4$ fluxes) = 11-13), horizontal and vertical lines are standard deviations. We decided to display production rates from 0-5 cm sediment depth as this depth turned out to be most important for sediment gas production (see sections 3.1.2 & 4.1.2). (c) Relative grain size distribution in sediment mesocosms.*





***Table 4:*** *Spearman's rank correlation coefficients for CH$_4$ and CO$_2$ fluxes (means of each sediment mesocosm) and different sediment characteristics.*

| | CH$_4$ flux | | CO$_2$ flux | |
|---|---|---|---|---|
| | rho | p | rho | p |
| **Clay** | 0.6477503 | 0.02275 | 0.6045669 | 0.03731 |
| **Silt** | 0.4966085 | *n.s.* | 0.3022835 | *n.s.* |
| **Sand** | -0.6477503 | 0.02275 | -0.6045669 | 0.03731 |
| **Fats, waxes, lipids** | -0.8333333 | 0.01538 | -0.3333333 | *n.s.* |
| **Phenols; humics** | -0.8333333 | 0.01538 | -0.3571429 | *n.s.* |
| **Aromates** | -0.5952381 | *n.s.* | -0.5238095 | *n.s.* |
| **Lignin** | -0.7863867 | 0.02063 | -0.3805097 | *n.s.* |
| **C/N** | -0.8809524 | 0.007242 | -0.3333333 | *n.s.* |
| **C (%)** | -0.7142857 | *n.s.* | -0.1904762 | *n.s.* |
| **CH$_4$ sediment stock change** | -0.2219706 | *n.s.* | 0.04985994 | *n.s.* |
| **CO$_2$ sediment stock change** | -0.04872526 | *n.s.* | -0.06414566 | *n.s.* |

Concentrations of dissolved CH$_4$ and CO$_2$ ($\Sigma$(CO$_2$, HCO$_3^-$, CO$_3^{2-}$)) in the sediment ranged from 3.97 to

129.45 mmol m$^{-3}$ (CH$_4$) and from 322.92 to 3811.36 mmol m$^{-3}$ (CO$_2$) respectively with lowest values found at site 3.125 and highest values at S.150. We did not see significant changes in CO$_2$ and CH$_4$ concentrations in the depth profile (see Fig. S4).

Observed changes of CH$_4$ and CO$_2$ concentration in the sediment of intact cores was overall very low and lay between 0.06 and 2.45 (CH$_4$) and 0.55 and 57.11 mmol d$^{-1}$ (CO$_2$) in the upper 5 cm of the

sediment. It was within the same order of magnitude as measured fluxes but did not correlate with the latter.

Grain sizes distribution differed between the four sites, with site 3.125 having the highest share of sand (21.6%) and lowest shares of silt and clay (35.68 and 42.73%) whereas the other sites were dominated by finer material and consisted of less than 5% of sandy components (see Fig. 8). CH$_4$ and CO$_2$ fluxes

were significantly correlated with clay and sand content, but not with silt (see Tab. 5).

Similar to incubations, CH$_4$ fluxes were significantly negative correlated with some FTIR peak ratios indicative of recalcitrant OM compounds as well as with C/N ratio but not with C content. CO$_2$ fluxes did not show any significant correlation with OM quality parameters at all (see Tab. 5).





## 4 Discussion

### 4.1 Sediment incubations

#### 4.1.1 Spatial variability of OM quality

In our study, we found higher C/N ratios in samples close to the lake shore and narrower ratios in the lake center indicating a predominant input of allochthonous (terrestrial) material at the shore and mainly sedimentation of autochthonous (aquatic) OM in the lake center (Meyers, 1994).

Besides C/N ratios, FTIR peak ratios revealed information about the OM composition and quality in Lake Windsborn. We found FTIR peak ratios for refractory components (lignin, aromatics, humics, phenols, and lipids) increasing towards the lake center, indicating increasing predominance of more recalcitrant OM with higher distance from the lake shore. At first sight, this was unexpected as allochthonous material is known to be richer in recalcitrant compounds like lignin or aromates compared to autochthonous biomass and therefore being effectively buried in lake sediments (Sobek et al., 2009). On the other hand, the internally produced material in the lake center has to bypass a deeper water column before reaching the sediment, meanwhile undergoing degradation processes and leading to more decomposed OM reaching the sediment (Meyers, 1994; Torres et al., 2011). The less decomposed OM close to the shore might further originate from labile aquatic plant substrates growing at the shoreline (Wetzel, 1992; Cole et al., 2007).

Not only on a spatial scale, but also with sediment depth, C/N ratios decreased and FTIR peak ratios increased, corresponding with a decrease in $CO_2$ and $CH_4$ production rates. While sediment age typically increases, OM quality and thus reactivity usually decreases with sediment depth, as recalcitrant compounds are not being mineralized completely but instead buried and preserved in the sediment (Avnimelech et al., 1984; Burdige, 2007; Sobek et al., 2009).

#### 4.1.2 Spatial variability and temperature dependency of $CO_2$ and $CH_4$ production rates

Our incubation experiment showed that both $CO_2$ and $CH_4$ production rates were highly variable within one lake of only small spatial extent. $CH_4$ production rates were within the range of formerly reported values from lakes in central Sweden, a reservoir in Brasil, and in sediments from Lakes Stechlin and Geneva in Germany and Switzerland (Duc et al., 2010; Fuchs et al., 2016; Grasset et al., 2018). $CO_2$ production rates were high compared to rates found in Lake Kinneret in Israel and in the Pantanal and Amazon regions in Brazil and exceeded reported values by a factor of 7 to 100 (Schwarz et al., 2008; Conrad et al., 2011). Compared to production rates after addition of fresh OM to lake sediment, our $CO_2$ and $CH_4$ production rates remained low (Grasset et al., 2018).

Both $CO_2$ and $CH_4$ production rates were higher in the topmost sediment layer compared to 5-10 cm sediment depth, suggesting that the first centimeters of the sediment play a major role for gas production as a consequence of temperature and microbial community distribution and changes in OM quality (as discussed below) (Falz et al., 1999; Sobek et al., 2009; Wilkinson et al., 2015).





In all samples, production rates were higher at the beginning of the experiment than towards the end.

While $CO_2$ production rates were highest right after the start and then constantly decreased until reaching a plateau around day 8, $CH_4$ production rates peaked after 3 to 8 days and then slowly decreased afterwards approaching a 1:1 $CO_2$:$CH_4$ production ratio. This typically observed 3-phase pattern (Yao et al., 1999) might e.g. be due to thermodynamic constraints, such as end product accumulation in the sample vials and subsequent inhibition of methanogenesis (Beer and Blodau, 2007; Blodau et al., 2011;

Bonaiuti et al., 2017). The observed delay in notable $CH_4$ production can be explained by the prevalent alternative EAs ($NO_3^-$, $Fe^{3+}$, $SO_4^{2-}$, humic substances) suppressing methanogenic activity (Lovley et al., 1996; Yao et al., 1999; Fenchel et al., 2012).

Similar to den Heyer and Kalff (1998), we found higher production rates in samples close to the lake shore in contrast to lower production rates in the center, suggesting that either the input rate of OM to

the sediment is higher and therefore fueling higher degradation rates or/and that the organic material at these sites is more labile and therefore more easily degradable for microorganisms. More details on that hypothesis will be given in the following.

The high measured variability in production rates showed that it is necessary to sample lake sediments from different locations in one lake, in order to understand potentials of $CO_2$ and $CH_4$ production

representative of the whole lake and that upscaling production rates based on single point measurements may be highly biased. Considering only sediment production rates, it seems that the in other studies observed spatial variability of $CO_2$ and $CH_4$ emissions from lakes (Schilder et al., 2013; Bastviken et al., 2015; Natchimuthu et al., 2016; Natchimuthu et al., 2017; Spafford and Risk, 2018) might to a large extent be controlled by sediment gas production. But still, as sediment production and emission patterns

have so far only been considered separately, important relations might have been neglected. In section 4.2, we therefore further discuss the relationships between $CO_2$ and $CH_4$ production and actual emissions and relate them to sediment properties.

Many studies revealed that higher temperatures lead to enhanced mineralization of OM due to higher microbial activity and thus increased production of $CH_4$ and $CO_2$ in sediments (den Heyer and Kalff,

1998; Sobek et al., 2009; Gudasz et al., 2015). Accordingly, we found that production rates of $CO_2$ were 2-times and of $CH_4$ 2 to 11-times higher with a temperature increase of 10°C. $Q_{10}$ values for $CO_2$ were within the range of earlier reported values by Liikanen et al. (2002) and Berström et al. (2010) whereas $Q_{10}$ values for $CH_4$ production were slightly higher than values found by Duc et al. (2010). The high observed range of $Q_{10}$-values, especially for $CH_4$, implies that responses of temperature increase might

not be homogeneously distributed within one single lake. The observed negative correlation between $Q_{10}$-values and OM quality indicators suggests that sites with more labile OM are more vulnerable to increasing temperatures in terms of $CH_4$ production and that at sites with more recalcitrant OM, the latter may limit the degradation processes. Aben et al. (2017) and Kiuru et al. (2018) showed that increasing temperatures caused higher $CO_2$ and $CH_4$ emissions from lakes. Assuming that gas




production in the sediment and efflux to the atmosphere are coupled, global warming may thus drastically enhance emissions from small lakes like Lake Windsborn, especially regarding the fact that these sediments warm much faster (max. bottom-water temperature in summer 2018: 26.65°C, surface water temperature at the same time: 27.14 °C) compared to deeper lakes which have a deeper water column and stable thermal summer stratification as buffer (Jankowski et al., 2006).

### 4.1.3 Influence of OM quality on $CO_2$ and $CH_4$ production rates

The amount and the quality of OM usually determines the degradability and therefore the production of end-products of anaerobic mineralization processes, $CO_2$ and $CH_4$.

Against our expectations and previous findings e.g. by Conrad et al. (2011) and Romeijn et al. (2019), C content was not correlated with $CO_2$ nor $CH_4$ production rates. As the production potential is obviously
not depending on the amount of C, we suggest that instead the OM quality might be the determining factor for $CO_2$ and $CH_4$ production.

Typically, OM of autochthonous origin fuels higher degradation rates than allochthonous OM (West et al., 2012; Grasset et al., 2018) as the latter is rich in compounds like lignin, cellulose and humic acids, being recalcitrant to decomposition and therefore being effectively buried in lake sediments (Sobek et
al., 2009). We however measured higher $CO_2$ and $CH_4$ production rates at sites close to the shore receiving higher inputs of allochthonous material compared to the center of the lake receiving mainly autochthonous OM as indicated by the C/N ratio. As a wider C/N ratio does not only indicate OM of terrestrial origin, but also implies that the OM is still less decomposed and therefore has a higher decomposition potential whereas a narrower C/N ratio is not only typical of aquatic OM but also
indicates a higher degradation state, higher production rates at sites closer to the shore seem to be reasonable (Malmer and Holm, 1984; Meyers, 1994; Kuhry and Vitt, 1996; Gudasz et al., 2015; Grasset et al., 2018). Anyway, the correlation between production rates and C/N ratio was not significant, implying that the C/N ratio might, due to before explained contradictions, not be a suitable proxy for the mineralization potential in our case. Instead, all FTIR peak ratios were significantly negative
correlated with $CO_2$ and $CH_4$ production rates, supporting the hypothesis that OM mineralization is highly dependent on the quality of the OM.

To summarize, the quality of OM is not necessarily depending on its origin and that OM of terrestrial origin is not necessarily more recalcitrant to degradation by aquatic microorganisms. Sites closer to the shore probably receive higher rates of terrestrial OM matter being less decomposed, whereas
autochthonous production in the lake center is overall low, due to the low nutrient status of the lake, and already undergoing degradation processes during sedimentation. This leads to more recalcitrant OM in the sediment. Earlier studies which found that allochthonous material is less decomposable compared to autochthonous matter were conducted in larger lakes, where the influence of the watershed is overall lower due to a lower perimeter to volume ratio. We hypothesize that in small and nutrient-poor lakes



the pattern might be reversed as the system adapts to overall low productivity and simultaneous high input of terrestrially produced OM.

### 4.1.4 Methanogenic pathways

To make methanogenesis a profitable reaction for microorganisms, theoretical thresholds of -10.6 kJ mol$^{-1}$ for the hydrogenotrophic and -12.8 kJ mol$^{-1}$ for the acetoclastic pathway have to be exceeded

(Hoehler et al., 2001). Except for a few samples at the beginning of the incubation, the theoretical thresholds were always exceeded, suggesting that both pathways contributed to CH$_4$ production. During the incubation, Gibb's free energy increased for the hydrogenotrophic pathway, implying an increasing contribution of that pathway to total CH$_4$ production, whereas it was opposite for the acetoclastic pathway. This suggests that labile organic polymers as precursors of acetate got depleted during the

incubation so that H$_2$ gained of importance as substrate for methanogenesis. But when considering H$_2$ and acetate concentrations, the picture gets more complicated. An overall increase of H$_2$ concentrations was observed during incubation, but average H$_2$ concentrations at t3 had a high standard deviation (see Fig. 5) because some samples showed a peak in H$_2$ concentrations at t3 instead of a constant increase. These patterns could hint at an imbalance of the system, where fermenting, syntrophic and methanogenic

bacteria were not yet equilibrated (Fey and Conrad, 2003). It is also remarkable, that the energy yield for acetoclastic methanogenesis decreased although acetate concentrations increased. We propose two different explanations for this finding: First, acetate is also used to reduce other EAs, which were present in higher concentrations at the beginning of the incubation, and therefore might have kept acetate concentrations low. Second, higher acetate concentrations do not necessarily mean more favorable

thermodynamic conditions due to end product accumulation. During degradation of carbohydrates, H$_2$ and acetate are being produced in a 2:1 ratio. We found acetate concentration exceeding H$_2$ concentrations by a factor of 10$^3$, suggesting that homoacetogenesis contributed significantly to acetate production in our incubations (Conrad, 1999).

Against our expectations, Gibb's free energy of acetoclastic methanogenesis was significantly lowest in

the center of the lake at t6 and was additionally positively correlated with C/N ratio. Considering that we found the lowest OM quality in the lake center and that low C/N ratio indicate OM of high recalcitrance whereas the acetoclastic pathway is attributed to predominate with high quality substrates, we would have expected a reverse pattern. Also, we found significant differences of energy yields between the sites, but could not relate them to OM quality. This implies that acetate as intermediate in

the anaerobic OM mineralization cascade is not the rate determining step, but rather is fermentation and that OM quality only explains a part of the variance in CO$_2$ and CH$_4$ production patterns.

It further has to be mentioned, that correlations with CO$_2$ and CH$_4$ production rates were only observed for H$_2$ concentrations at the end of the incubation, but not with acetate. This finding emphasizes the hypothesis that the system was not in balance from the beginning of the incubation, and that fermentation

seems to be the rate determining step for OM mineralization.





### 4.1.5 Alternative electron acceptors

Average $EAC_{OM}$ measured in our experiment ($302.8 \pm 124.6$ µmol $e^-$ $gC^{-1}$) was slightly lower but in the same order of magnitude compared to values found by Lau et al. (2015) and Gao et al. (2019) in peat soils and other lake sediments respectively. Given that C contents in the study by Gao et al. (2019) were

much higher than in our study (46.2 – 49.4 %, Agethen and Knorr (2018)) and that Lau et al. (2015) fully oxidized their samples (C content 11.0 – 27.4 %), our measured capacities seem reasonable.

We found both organic and inorganic EAs decreasing during the experiment, indicating that EAs are being depleted with time as they are being reduced by microorganisms.

Sites with higher $EAC_{tot}$ had lower $CH_4$ production rates and vice versa as increased concentration of

oxidized humic substances would suppress methanogenic activity, supporting the findings of Agethen et al. (2018) and Gao et al. (2019). Although the relationship between EAC and $CH_4$ production was – probably due to the low number of samples - not significant, a clear trend was observable. Additionally, relationships were stronger with $EAC_{tot}$ that with $EAC_{inorg}$ or $EAC_{org}$ only, suggesting that humic substances play a crucial, but variable role as terminal electron acceptors and should not be neglected

when estimating the contribution of EAs to methanogenic activity in lake sediments rich in OM (~30%) (Lau et al., 2016). The spatial variability of measured EAC and C content in the samples shows that substrate composition is critical when evaluating the contribution of different EAs and $CH_4$ production patterns.

### 4.2 Comparison of production rates, fluxes, and sediment parameters

Comparing the order of magnitude of potential production rates from incubations and fluxes from intact sediment cores, results were in good accordance with each other. While $CO_2$ production rates were overall something below measured fluxes, $CH_4$ production rates were higher than $CH_4$ fluxes. The differences in $CO_2$ production rates and fluxes can be explained by the influence of the water column: $CO_2$ is not only produced in the sediment, but also in the water column, e.g. by the degradation of OM

or zooplankton respiration (Kling et al., 1992) and can therefore contribute to water-atmosphere fluxes. The differences concerning $CH_4$ could be explained by several approaches. First, preparing incubations leads to the homogenization of the sediment and higher surface area, and therefore even distribution of substrates and microorganisms which is then better available for the latter (Hoehler et al., 2001). Secondly, sediment fluxes from intact sediment cores may be low compared to homogenized

incubations as we did not observe significant increases in the depth profile of sediment cores and suggest that strong changes in sediment gas concentrations which are driving fluxes are likely to occur only in the upper mm to cm. Additionally, the preparation of incubations reduces thermodynamic constraints which usually exist in intact profiles due to end product accumulation (Blodau et al., 2011; Bonaiuti et al., 2017). Thirdly, not all $CH_4$ produced in and emitted by the sediment will reach the water-atmosphere

interface because of consumption processes in the oxygenated water column that has to be passed.



Through $CH_4$ oxidation in the water column, amounts of $CH_4$ emitted from lakes to the atmosphere can be substantially reduced by 51 to 100 % (Bastviken et al., 2002; Bastviken et al., 2008).

Interestingly, spatial patterns of $CO_2$ and $CH_4$ emissions could not be well reproduced from sediment incubations. Although highest $CO_2$ emissions were observed at site S.150, we found lowest $CO_2$ production rates there. Additionally, we did not see any correlations between sediment OM quality and $CO_2$ fluxes, suggesting that $CO_2$ production processes in the sediment overlying water column play a much more important role than in the sediment itself. Moreover $CH_4$ production rates at sites 1.50 and 3.125 differed significantly from each other while this was not the case when comparing fluxes. We assume that the different patterns of $CH_4$ fluxes and production can be explained by grain size distribution and thus physical sediment properties. Sediment grain size determines sediment pore size which is essential for the evolution of methane bubbles in the sediment (Boudreau et al., 2005). Ebullition is the major component of $CH_4$ fluxes in shallow waters as it can efficiently bypass oxidation in the water column (Bastviken et al., 2004; Wik et al., 2013; Natchimuthu et al., 2016). We found ebullition supporting significantly to total $CH_4$ fluxes in two of our four sediment mesocosms, whereas sites with higher shares of sand exhibited less ebullitive fluxes confirming the findings of Liu et al. (2016) and (2018). We further found OM quality being partly significantly correlated to $CH_4$ fluxes, but to a lesser extent than to $CH_4$ production. These findings suggest that grain size distribution is besides OM quality a main driver of spatial $CH_4$ flux patterns in mesocosms. On the other hand, when preparing incubations, the physical sediment structure is destroyed, so that OM quality becomes the major controlling factor for gas production. Although we also found OM quality being significantly correlated to mesocosm fluxes, we suppose that the combination of physical characteristics and sediment OM quality can sufficiently explain $CH_4$ emission patterns from lakes.

The missing link between sediment gas storage and both diffusive and ebullitive emissions could be explained by our sampling procedure. While fluxes were partly dominated by bubbles, these bubbles cannot be captured in the sediment by silicon samplers. Silicon samplers rely on the slow diffusion of gas between the pore gas and the sampler. Bubbles however may form on shorter time-scales and could quickly being released to the water-column. Thus, concentrations in the bulk sediment stay similar while the released bubbles cause high fluxes. Moreover, the resolution of our gas samplers was probably not high enough, not covering the uppermost centimeters of the sediment which we assume to be crucial for diffusive fluxes.

## 5 Conclusion

Our experiment showed that there exists a significant spatial variability of $CO_2$ and $CH_4$ production in the sediment of a small and shallow lake and that it is therefore not possible to upscale sediment production rates from single point measurements. We further proved that especially $CH_4$ production is strongly depending on temperature, and that the extent of temperature dependence might differ within one lake, especially between littoral and central parts due to different OM quality. Small lakes seem to



be very vulnerable to temperature increases in terms of accelerated C cycling which implies they might become larger sources of $CO_2$ and $CH_4$ emissions under climate change scenarios. Still, they might not react homogeneously, but we could unroll that sediments in shallower parts might react more sensitively

due to lower water columns and input of reactive allochthonous OM, which is particularly important for oligotrophic lakes without significant contribution of autochthonous primary production. The strong negative correlations we found between recalcitrant OM compounds and $CO_2$ and $CH_4$ production rates proved evidence that OM quality plays a major role in controlling the mineralization of anoxic lake sediment and can be used to explain and predict within-lake patterns of production rates. Both

hydrogenotrophic and acetoclastic methanogenesis were feasible during incubations, but we could neither clearly refer observed patterns to OM quality nor electron exchange capacities, and therefore suggest that both $H_2$ and acetate were not limited as substrates but that fermentation was the rate-determining step of OM mineralization. More than organic or inorganic EAs alone, $EAC_{tot}$ could explain observed variabilities in $CH_4$ production, implying that also organic EAs play a major role in constantly

anoxic lake sediments with high C contents.

However, measured production rates were only partly reflected in $CO_2$ and $CH_4$ flux patterns obtained from whole core incubations. We suggest that this is due to the destruction of physical sediment parameters in the incubations (e.g. pore size) which are crucial for the evolution of $CH_4$ bubbles in the sediment and due to the lack of thermodynamic and gas transport constraints that exist in intact anoxic

sediment profiles. Further, measuring production rates neglects the importance of the water column as a sink of sediment generated $CH_4$ through oxidation and source of $CO_2$ through degradation and respiration processes which might then cause discrepancies between observed production and flux rates.

So far, direct flux measurements of $CO_2$ and $CH_4$ between water and the atmosphere seem to be the most accurate way for determining the magnitude and variability of emissions from lakes, but still our study

contributes substantially to the understanding of controls on $CO_2$ and $CH_4$ production processes in lake sediments.



## 6 Data availability

The underlying research data of this study can be found in the Supplement.

## 7 Author contribution

LP, MS, SS, NP and KHK designed the study. NP and SS carried out the experiments and sample analyses with the help of LP and MS. LP performed data analyses and wrote the manuscript with the help of NP, SS, KHK and GB.

## 8 Competing interests

The authors declare that they have no conflict of interest.

## 9 Acknowledgements

We thank the technicians of the laboratory of the institute of landscape ecology Ulrike Berning-Mader, Madeleine Supper, Viktoria Ratachin and Melanie Tappe for sample measurements. We further thank Rebecca Pabst, Christina Hackmann, Fabian Lange, Isabelle Rieke, Friederike Ding, Victoria Tietz
and Anja Radermacher for their assistance during laboratory work and Stephan Glatzel for grain size analyses. This project was funded by the German Research Foundation DFG (BL 563/25-1). We acknowledge support from the Open Access Publication Fund of the University of Muenster.



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
