# Peer review of "Organic matter and sediment properties determine in-lake variability of sediment CO2 and CH4 production and emissions of a small and shallow lake"

_Biogeosciences, 2019_

## Referee Comment (RC1) · Anonymous Referee #1 · 23 Sep 2019

The study measured transects of sediment characteristics at two depths across a shallow eutrophic lake. Besides chemical and physical variables, the authors measured production rates of CH4 and total CO2 and correlated them to various sediment variables, in particular to FTIR spectra resolving polysacchcarids, lignin, humic acids, phenols and aliphatics, other aromatics and fats, waxes, lipids; and to the magnitude of electron accepting and donating capacities. They further measured CH4 and CO2 fluxes in the centre of the lake and compared these rates to the production rates. The study provided a solid data base and many interesting correlation analyses. Highlights

are the negative correlation of sediment gas production rates to recalcitrant organic compounds (fats, humics) and electron accepting capacities. Furthermore there was no obvious correlation between gas production rates and emission fluxes. These were to my opinion the most interesting results that warrant reporting, while observations with respect to the influence of temperature and the thermodynamics of CH4 production pathways were rather trivial, as they are well known from the literature.

However, I noticed several points that should be addressed by the authors before the ms is accepted. These comments are also found in the pdf supplemental file.

Major points:

1.L.79-81: The Gibbs free energies given in the ms are either not found in the quoted literature (Whiticar 1999) or are different (Conrad 1999). I assume the reason is that they were calculated using energies of formation for gases in dissolved rather than in gaseous state. This would be consistent with the Nernst equations mentioned later (L.250) also probably using gas concentrations rather than partial pressures. However, the authors should clarify the procedures.

2.L.187-200: There are no isotopic data reported, therefore the description of IRMS methodology is not necessary.

3.Table 2: L.411-412 mentions strong FTIR absorption features of polysaccharides. However, this compound class is not listed in the Table.

4.L.422-425. This is an overview of measured rates. However, the numbers seem to be slightly different from those shown in Fig.2. Although there is probably a reasonable explanation for these differences, I found it confusing. In fact I would be happy just looking at the data in the figure without reading the text. However, one could mention that the rates decreased from the shore to the centre, since this point is later relevant in the Discussion.

5.L.477-487: Here applies the same as in point 4. The data in the text seem to slightly

different from those seen in Fig. 7.

6.L.507: Again the data in the text seem to slightly different from those seen in Fig. 8.

7.The discussion is too wordy and should be focused to the really novel results. I also recommend a different structure for the Discussion. I think it is not ideal having individual chapters on spatial variability of OM quality, spatial variability of $CO_2$ and $CH_4$ production rates, and influence of OM quality on gas production, since such structure results in too much repetition and also is not very suitable for explaining gas production rates on the basis of OM quality.

9.The discussion on temperature effects can be much shorter, since it is rather well reported in the literature.

10. The discussion of methanogenic pathways (L.648-680) is not really relevant, since the data just show that both methanogenic pathways were exergonic and thus, could well operate. Everything else is speculation and not relevant. The magnitude of the Gibbs free energy does not allow to conclude whether the one pathway is more prevalent than the other. One could however discuss the correlation of the concentrations of H2 and acetate, and the respective Delta G, with sediment OM quality, since correlations were reported in the Results.

11. The discussion of alternative electron acceptors (L.682-698) is rather short. The authors only discuss correlations. They miss the chance to discuss stoichiometric relations of reduced EAC with the amounts of $CO_2$ production. Although such mass comparisons apparently have recently been done by other members of the Knorr group (Gao et al. 2019), they would also be interesting for this particular lake. I have the impression that the magnitudes of reducible EACs might explain the $CO_2$ production in the beginning of the incubations, when rates of $CO_2$ production were larger than those of $CH_4$ production, while methanogenic decomposition of OM should result in equal rates. I wonder why this point is not addressed.

12. I noticed that lake sediments were anoxically preincubated for either one week (L:178) or 50 days (L.331). Please clarify! Anyway, the preincubation might have depleted most of the reducible iron and sulfur compounds. This may be the explanation for the low values of EACinorg (Fig. 7), but is not discussed.

Minor: 1.L.28: what means 'sufficiently' ? rho=0.65 is sufficient? Would rho=0.6 also be sufficient. Is there an objective criterion for sufficiency? 2.L.30-32. I cannot follow the argument of this sentence. I suggest rephrasing. 3.L.67: cellulose is also a polysaccharide. I suggest rephrasing. 4.L.83: The Delta G-zero of hydrogenotrophic methanogenesis is more negative than of aceticlastic methanogenesis. Therefore the acetoclastic pathway is less (not more) energetically favorable. 5.L.91: EAC has not yet been defined. Please check also for other ebbreviations. 6.L.107-109: The '4' in CH4 as superscript 7.L.168: 12 locations; please harmonize with the 13 sampling sites mentioned in the legend of Fig.1. 8.L.207: 'relative abundance' compared to what? 9.L.268: EAC/EDC: I think you mean EAC & EDC rather than the ratio between both. I found similar possible confusions at many places in the text (e.g., L.293, L.369, 370, 383 and in the labels of Fig. 7. Please check carefully. 10.L.299. The reference Tamura et al. (1974) only describes the analysis of Fe(II) (albeit in the presence of Fe(III)). How was Fe(III) analyzed? 11. L.477-479: I cannot follow this sentence. Also compare major point 6 above. Please also note, that Fig. 7 is not mentioned in the text, and that Figure number should be exchanged with that of Fig. 6, since Fig.6 is reported later in the text than Fig. 7. 12.Table 3: Showing the time line as t0, t1, t2 etc. is awkward, since one has to consult the explanation in the methods section. I suggest listing the actual time points, i.e. 0, 1, 3 etc. days. 13. Table 4: The numbers in the table show too many decimal positions. Please report only those that are significant. In fact, at numerous places in the text numbers seem to show non-significant decimal positions. Please check and correct. 14.L.535, 538: Should be Table 4 rather Table 5. 15. References. Some of the references use capital letters for the titles.

Please also note the supplement to this comment:

https://www.biogeosciences-discuss.net/bg-2019-284/bg-2019-284-RC1-supplement.pdf

---

## Referee Comment (RC2) · Anonymous Referee #2 · 12 Apr 2020

Review of the manuscript, "Organic matter and sediment properties determine in-lake variability of sediment CO2 and CH4 production and emissions of a small and shallow lake' by L.S.E. Praetzel et al.

This manuscript highlights large variations in production and fluxes of two of the most important greenhouse gases (CO2 and CH4) in a small and shallow lake located in Southwest Germany. The authors present a large data set and their interpretations are by and large sound and interesting. The manuscript certainly deserve publication, but with substantial modifications. I got involved with the review process quite late and

therefore had the benefit of going through the other review, which is available online. I fully agree with the other referee's evaluation and only add here a few more comments that hopefully will be helpful in improving the presentation.

General

The manuscript needs careful line editing to take care of non-idiomatic English. An example is the frequent usage of wrong tenses (e.g. in line 60: "is mainly depending on" rather than "mainly depends on"). The authors may seek help from a native English speaker for this purpose. I have pointed out a few instances below, but these are by no means exhaustive. I must also concede that I am not a native English speaker!

Specific

Lines 14-15: Change "... to the atmosphere, following recent studies this is particularly the case for small and shallow lakes." to "... to the atmosphere; recent studies have shown that this is particularly the case for small and shallow lakes."

Line 16: Delete "yet" and "thus".

Lines 21-22: Change "... were significantly negative (p<0.05, rho<-0.6) correlated" to "... exhibited significant negative correlation (p<0.05, rho<-0.6)". Please make similar changes elsewhere.

Lines 32-34: The last sentence states the obvious. Who has suggested such a "re-placement"?

Line 52: Change "has been" to "have been" (here majority is plural), and remove "is".

Line 56: Remove hyphen between "in" and "lake".

Lines 58-59: Why is it crucial? Your results show that it is not.

Line 64: Also its origin (e.g. lignin).

Line 74: Remove "being".

Line 82: As also pointed out by the other referee a more negative deltaG change would make R2 thermodynamically more favourable.

Lines 86-87: Change "are attributed to" to "may arise from".

Line 90: Change "remain" to "remains".

Lines 92-93 and elsewhere: As also pointed out by the other referee please define each abbreviation when you use it the first time and maintain consistency.

Line 96: Why "to a small extent"? In such shallow systems wind-driven turbulence could disturb the sediments.

Lines 97-98: Add "penetration" after "oxygen" and remove "in our case". What do you mean by "perennial circulation".

Line 99: Please use present indefinite tense, not present continuous.

Line 103: "other" sediment properties?

Line 108: Change "is accountable" by "accounts".

Lines 110-114: Please rephrase this sentence.

Line 119: Did you actually investigate "connected productions patterns to OM"?

Line 121 and elsewhere: I am not sure if these experiments can be termed as "meso-cosm". These were incubations of cores in the lab. Line 125: Change "hypothesize" to "hypothesized". Line 137: Change "blast" to "blasted". Line 138: Change "arose" to "formed". Figure 1 captions: Technically the depth categories are wrong. For example by <150 cm, you imply depths between 125 and 150 cm, but 20 cm is also <150 cm. This should be clarified (e.g. 125 indicates 100cm<depth≤125 cm).

Table 1, caption: Analytical procedued do not have to be mentioned here; they should be described in methodology section.

Line 167: Change "at three occasions" to "on three occasions".

Line 169: Why randomly? It should be selectively based on a reason.

Line 172: Add "respectively" at the end of the sentence.

Line 176: Change "added with" to "containing".

Line 180: Change "stored" to "maintained".

Line 192: Referring to a comment by the other reviewer, I note that some isotope data are presented in Table 2 (not for sulphur though), although not at all discussed in the text. It is not clear whether the sample was decalcified. Also what was the reproducibility of measurements? In fact the precision of analysis is not given for any parameter.

Line 201: Change "Therefore" to "For this purpose".

Line 253: Something missing in the sentence.

Line 267: Change "analyzed for" to "measured".

Lines 274, 291: Change "measured" to "analyzed". (Note samples are analyzed, parameters are measured).

Line 301: Change "Therefore" to "For this purpose".

Line 313: These are lab experiments, NOT mesocosms!

Line 331: Change "conducted" to "made".

Line 361: There is no other way to quantify inputs is ebullition?

Line 408: Change "nor" to "or".

Fig. 2: Figure difficult to digest. I could not follow "Different letters indicate significant differences between these sites." What do letters "a"-"d" mean? Am I missing anything?

Line 483: Change "by averagely" to "on a average by".

Lines 490, 493, 499: See earlier comments on Lines 21-22.

Line 508 and elsewhere: I believe sediment ebbulition in inferred from k>o. I am not sure. Was there any bioturbation that could increase the emission?

Table 4: "n.s." presumably means not significant (p<0.05). Is is mentioned somewhere? Significance also depends on the number of values that are not given.

Lines 528: Change "concentration" to "concentrations" and "was" to "were".

Line 536: Change "were significantly negative correlated" to "showed significant negative correlation"

Line 542: What do you mean by "narrower"? lower?

Lines 545-556: Authors have emphasized on C/N ratio. They have observed increase in C/N with depth in the inner part of the lake. C/N ratio may not be a very efficient parameter to characterize organic source owing to rapid remobilization of nitrogen as well as reabsorption of ammonium on particulates. The paragraph 405 "The C content in the samples was between 2.15 and 33.16% with lowest values at site 3.50 and highest at site 1.50. C/N ratios ranged from 10.97 at site 1.150 to 19.06 at site 3.100. Neither C content nor C/N ratio showed significant changes with sediment nor lake depth, but C/N ratio was significantly higher in samples taken close to the shore (50) than in samples from the lake center (150) (p < 0.01)." is very confusing. A graph showing distribution of C/N ratio across the horizontal length of the lake would suitable to comprehend the results better.

Line 551-552: I do not believe in shallow depths it matters.

Line 559: Change "buried" to "getting buried"

Line 571: Change "role for" to "role in"

Line 578: Change "e.g." to "among other things"

Line 587: Change "in the following" to "below"

Line 591: Remove "the" before "in other studies"

Lines 612-614: Laborious sentence. All you are saying is that such shallow depths do not get thermally stratified in summer.

Lines 619-620: Change the tense to present indefinite.

Line 627-630: What do you mean by "wider" and "narrower"? I do not follow this sentence.

Line 637: Change the tense to present indefinite.

Line 654: But the acetate concentration increased!

Line 655: Remove "of" before "importance"

Line 671: OM quality is not quantified so instead of low you should perhaps use poor.

Line 673: Change "of energy" to "in energy"

Line 675: Change "... acetate, but rather is fermentation" to " ... acetate. Instead fermentation may be rate limiting"

Line 677: Bring "Further" before "it".

Line 678: Change "finding emphasizes" to "supports"

Line 692: If the relationship was insignificant the trend cannot be "clear".

Line 693: Not at all clear, and so is the following conclusion. I find this whole paragraph speculative.

Line 702: Change "something" to "somewhat"

Line 706: Change "approaches" to "factors"

Line 730: Authors attempt to correlate ebullition with grain size. They believe that

higher sand content leads to lesser ebullition. Which is highly unlikely since ebullition depends on permeability of sediments and not porosity. Sand always has higher permeability than silt and clay although lesser porosity. You need to elaborate your concept with more clarity

Line 747: Change "experiment" to "results"

Line 749: Remove "especially"

Line 753: Change "vulnerable" to "sensitive"

Line 754: Change "unroll" to "expect"

Line 755: Change "lower water columns" to "shallow depths"

Line 761: Change "refer" to "attribute"

Lines 764-765: Then why do you not find strong relationship between methane production and (EACorg)?

Line 770: Measuring "production rate" does not neglect water column processes, interpretation of these data alone would.

---

## Author Response (AR1)

**Authors' response to referee's comments (RC1) on the manuscript, "Organic matter and sediment properties determine in-lake variability of sediment CO2 and CH4 production and emissions of a small and shallow lake" by L.S.E. Praetzel et al.**

Dear reviewer,

thank you very much for your comments and suggested improvements on our manuscript "Organic matter and sediment properties determine in-lake variability of sediment CO2 and CH4 production and emissions of a small and shallow lake". We are sure that consideration of your raised points and ideas will considerably improve the manuscript and therefore we will especially focus on revising the discussions part. Please find our responses to each of your comments below, they will be structured as follows:

(1) comments from referee, (2) author's response, (3) author's changes in manuscript.

We hope to be able to answer your open questions and to adequately include your suggestions.

Kind regards,
Leandra Praetzel & Co-Authors

**Major points:**

(1) 1.L.79-81: The Gibbs free energies given in the ms are either not found in the quoted literature (Whiticar 1999) or are different (Conrad 1999). I assume the reason is that they were calculated using energies of formation for gases in dissolved rather than in gaseous state. This would be consistent with the Nernst equations mentioned later (L.250) also probably using gas concentrations rather than partial pressures. However, the authors should clarify the procedures.

(2) Gibb's free energies were calculated using formation energies Gf0 for the elements involved in the reactions in aqueous state. Used Gf0 values are reported by Stumm & Morgan 1996 and Nordstrom & Munoz 1994. In the Nernst equation, dissolved and gaseous concentrations in incubation vials were used for calculations. This will be mentioned in the revised manuscript.

We will include the following explanation on calculation of delta Gr0 in the revised manuscript:

(3) *"Values for Δ Gr0 are calculated from standard formation energies Δ Gf0 at 25°C in aqueous state listed in Stumm and Morgan (1995) and Nordstrom and Munoz (1994)."*

(1) 2.L.187-200: There are no isotopic data reported, therefore the description of IRMS methodology is not necessary.

(2) delta 13C and delta 15N values are reported in Table 2 and will be discussed in more detail in the results section of the revised manuscript. IRMS was also used to determine mass contents of C and N. The authors therefore prefer to leave the methodology description in the manuscript.

(3) *"C and N isotopic values did not vary much between sites and were on average -27.6 ‰ and -0.9 ‰ respectively with only one outlier for δ15N at site 3.50 (-6.3 ‰)."*

*"Neither CO2 nor CH4 production rates were exhibited significant correlation with C content or, C/N ratio, δ13C or δ15N,…"*

(1) 3. Table 2: L.411-412 mentions strong FTIR absorption features of polysaccharides. However, this compound class is not listed in the Table.

(2) Absorption maxima for polysaccharides will be listed in Table 2 of the revised manuscript.

(1) 4.L.422-425. This is an overview of measured rates. However, the numbers seem to be slightly different from those shown in Fig.2. Although there is probably a reasonable explanation for these differences, I found it confusing. In fact I would be happy just looking at the data in the figure without reading the text. However, one could mention that the rates decreased from the shore to the centre, since this point is later relevant in the Discussion.

(2) We double-checked the accordance of production rates stated in the text and displayed in Figure 2. We confirm that values in the text are the same as in the figure. We note that production rates in the text are mean values, whereas bold lines of boxplots are median values. We suppose that this fact might have caused the perceived discrepancy.

Moreover, we will simplify the figure by removing data from 5-10 cm depth and instead include this data in the supporting information.

We will further add a sentence to emphasize that production rates increased from the center to the shore as follows:

(3) "Overall, production rates decreased from the shore to the center of the lake."

(1) 5.L.477-487: Here applies the same as in point 4. The data in the text seem to slightly different from those seen in Fig. 7.

(2) See also answer to point 4. We double-checked the data and confirm that the values in text and figure are the same. Please note that the stated values in the text are mean values +/- standard deviations of triplicate measurements at the respective sites where minima and maxima were measured whereas boxplots in the figure represent the whole dataset at each sampling date.

(1) 6.L.507: Again the data in the text seem to slightly different from those seen in Fig. 8.

(2) We will adjust the mentioned values in the text according to the displayed flux rates in Fig. 8 as follows:

(3) "$CO_2$ and $CH_4$ fluxes measured from intact sediment core incubations ranged from 10.8 ± 4.4 to 17.9 ± 2.0 and 0.02 ± 0.01 to 1.5 ± 2.6 mmol $m^{-2}$ $d^{-1}$ respectively."

(1) 7. The discussion is too wordy and should be focused to the really novel results. I also recommend a different structure for the Discussion. I think it is not ideal having individual chapters on spatial variability of OM quality, spatial variability of CO2 and CH4 production rates, and influence of OM quality on gas production, since such structure results in too much repetition and also is not very suitable for explaining gas production rates on the basis of OM quality.

(2) We will shorten the discussion and focus on novel results in the revised manuscript. The chapters 4.1.1, 4.1.2 and 4.1.3 will be summarized as one, entitled "4.1.1 Variability of CO2 and CH4

production rates and influence of OM". Like this, we will considerably shorten the paragraph and thus avoid repetitions.

(1) 9. The discussion on temperature effects can be much shorter, since it is rather well reported in the literature.

(2) In the discussion on temperature effects, we emphasize the sensitivity of small lakes to temperature changes compared to large lakes. We will shorten that section in the revised manuscript but prefer to leave that statement in the discussion, as this strong and variable temperature effect on production rates has, to our knowledge, not been shown before for small and shallow lakes.

*(3) "In accordance with previous studies (den Heyer and Kalff, 1998; Sobek et al., 2009; Gudasz et al., 2015), we found that with a temperature increase of 10°C, production rates of CO2 doubled and those of CH4 were 2 to 11 times higher. Q10-values for CO2 were thus within the range of earlier reported values by Liikanen et al. (2002) and Berström et al. (2010), whereas Q10-values for CH4 production were slightly higher than values found by Duc et al. (2010). The large observed range of Q10-values, especially for CH4, implies that responses to temperature increases might not be homogeneously distributed within a lake. We point out that sediment CH4 production is more sensitive to increasing temperatures compared to CO2 production and that this leads to a stronger feedback on global warming when considering the higher global warming potential of CH4 compared to CO2 (Marotta et al., 2014). The observed negative correlation between Q10-values and FTIR peak ratios further suggests that sites with more labile OM are more susceptive to increasing temperatures in terms of CH4 production, whereas at sites with more recalcitrant OM, this recalcitrant OM may limit the degradation processes. We therefore assume that sediment greenhouse gas production in small and shallow lakes might in the course of global warming increase to a larger extent than in deeper lakes, as shallow waters, compared to deeper lakes, do not get thermally stratified in summer and therefore shallow sediments warm much faster (Jankowski et al., 2006)."*

(1) 10. The discussion of methanogenic pathways (L.648-680) is not really relevant, since the data just show that both methanogenic pathways were exergonic and thus, could well operate. Everything else is speculation and not relevant. The magnitude of the Gibbs free energy does not allow to conclude whether the one pathway is more prevalent than the other. One could however discuss the correlation of the concentrations of H2 and acetate, and the respective Delta G, with sediment OM quality, since correlations were reported in the Results.

(2) We will shorten the discussion of methanogenic pathways and increase the focus on the reported relationships between Gibb's free energies and OM quality in the revised manuscript.

*(3) "[…] suggesting that both pathways could contribute to CH4 production during the whole experiment. Still, this approach does not allow to evaluate which of the pathways predominates."*

*"[…] we would have expected a reverse pattern. Concomitantly, Gibb's free energy of hydrogenotrophic methanogenesis exhibited significant positive correlations with some FTIR peak ratios, although we expected that a high abundance of recalcitrant OM compounds would make hydrogenotrophic methanogenesis more feasible (Miyajima et al., 1997). Acetate and H2 concentrations on the other hand, both exhibited significant negative correlations with some FTIR peak ratios. While this seemed reasonable for acetate concentrations (less acetate available in strongly decomposed OM), this result again proved to be against our expectations in terms of H2 concentrations. One reason for these ambiguous findings might be that the system was not in a*

*steady state in terms of thermodynamic conditions. Similarly, long time scales of experiments still not reaching such steady state have also been observed earlier for peat columns (Bonaiuti et al., 2017)."*

(1) 11. The discussion of alternative electron acceptors (L.682-698) is rather short. The authors only discuss correlations. They miss the chance to discuss stoichiometric relations of reduced EAC with the amounts of $CO_2$ production. Although such mass comparisons apparently have recently been done by other members of the Knorr group (Gao et al. 2019), they would also be interesting for this particular lake. I have the impression that the magnitudes of reducible EACs might explain the $CO_2$ production in the beginning of the incubations, when rates of $CO_2$ production were larger than those of $CH_4$ production, while methanogenic decomposition of OM should result in equal rates. I wonder why this point is not addressed.

(2) In the revised manuscript, we will, besides observed correlations, also discuss the stochiometric relationships between measured EAs and $CO_2$ production: We find that calculated $CO_2$ production from prevalent EAs is lower than the measured $CO_2$ production (see. Fig.). We propose this is due to unknown consumed EAs during the incubation that we did not capture; most likely iron in the solid phase. We indeed measured solid phase iron, but are not able to make statements about its speciation. But starting from the total Fe concentration in the solid phase (2-3%), we suppose that this is high enough to explain the missing EAs to reach a 1:1 ratio of measured $CO_2$ production and calculated EA turnover.

[Figure]

(1) 12. I noticed that lake sediments were anoxically preincubated for either one week (L:178) or 50 days (L.331). Please clarify! Anyway, the preincubation might have depleted most of the reducible iron and sulfur compounds. This may be the explanation for the low values of EACinorg (Fig. 7), but is not discussed.

(2) The incubations were pre-incubated for one week, data from the sediment mesocosms was only used for analysis after 50 days of deployment in the climate chamber, in order the leave the cores adapt to laboratory conditions. We verified steady state conditions via the gas concentration in the sediment profile and observed constant conditions (i.e. no increase of concentrations) after 50 days. We will give a more detailed explanation on the procedure concerning sediment mesocosms in in the revised manuscript:

*(3) "For statistical analyses and discussion, we only used measurements that were made >50 days after the deployment of the intact sediment core incubations in the climate chamber. This was done in order to ensure the system had adapted to experimental conditions and had reached a steady state. Steady state conditions were indicated by quasi-constant $CO_2$ and $CH_4$ concentrations in the sediment."*

(2) We will additionally change the subtitle of the sections to clarify the difference between incubations and sediment mesocosms:

*(3)      2.3 Intact sediment core incubations*

*2.3.1 $CO_2$ and $CH_4$ fluxes*

*2.3.2 Sediment gas stock change*

(2) Further, as suggested by the other reviewer, we will rename the sediment incubations to "slurry incubations" and the sediment mesocosms to "intact sediment core incubations".

(2) The low EACinorg values due to 1 week of preincubation will be discussed in the revised manuscript.

*(3) "Nevertheless, both absolute $EAC_{inorg}$ values as well as relative changes were very low, which might have been caused by the one-week preincubation, where most of the reducible inorganic compounds might have already been depleted."*

**Minor:**

(1) 1.L.28: what means 'sufficiently' ? rho=0.65 is sufficient? Would rho=0.6 also be sufficient. Is there an objective criterion for sufficiency?

(2) The word sufficiently will be removed from the text.

(1) 2.L.30-32. I cannot follow the argument of this sentence. I suggest rephrasing.

(2) The sentence will be rephrased in the revised manuscript as follows:

*(3) "Our results show that within a small lake, CO2 and CH4 production show significant spatial variability, which is mainly driven by spatial differences in the degradability of the sediment OM."*

(1) 3.L.67: cellulose is also a polysaccharide. I suggest rephrasing.

(2) Cellulose will be replaced by natural organic matter in the revised manuscript.

(1) 4.L.83: The Delta G-zero of hydrogenotrophic methanogenesis is more negative than of aceticlastic methanogenesis. Therefore the acetoclastic pathway is less (not more) energetically favorable.

(2) This will be corrected in the revised manuscript.

(1) 5.L.91: EAC has not yet been defined. Please check also for other abbreviations.

(2) This will be corrected in the revised manuscript. We will also check for other abbreviations and make sure that they will be defined when being used the first time and that afterwards, abbreviations will be used consistently throughout the whole manuscript.

(1) 6.L.107-109: The '4' in CH4 as superscript

(2) This will be corrected in the revised manuscript.

(1) 7.L.168: 12 locations; please harmonize with the 13 sampling sites mentioned in the legend of Fig.1.

(2) Samples for incubations were taken from 12 sampling site, samples for sediment mesocosms were taken from 4 sites, whereas one of the mesocosms (S.150) represents the three sites of 150 cm depth from the incubations (1.150, 2.150 and 3.150) so that there are in total 13 sampling sites. This will be explained in more detail in the caption of Fig. 1 as follows:

*(3) "Location of the study area and 13 sampling sites within Lake Windsborn. Rhombus: Sampling sites for slurry incubations and intact sediment core incubations, circles: sites for slurry incubations only, asterisk: site for intact sediment core incubation only (reference for 1.150, 2.150 and 3.150)."*

(1) 8.L.207: 'relative abundance' compared to what?

(2) FTIR absorption peaks show relative abundances of the corresponding functional moieties in a single sample compared to another moiety. We therefore calculate peak ratios normalized to the polysaccharide peak. The sentence will be changed as follows in the revised manuscript:

*(3) "Distinct peaks at specific wavelengths were assigned to functional groups according to Artz et al. (2008) and normalized to the peak intensity at 1031 - 1035 cm-1 (indicative of polysaccharides) in order to obtain inter-comparable peak-ratios of functional moieties in all samples as FTIR spectra only provide information about the relative abundance of certain functional moieties in one sample."*

(1) 9.L.268: EAC/EDC: I think you mean EAC & EDC rather than the ratio between both. I found similar possible confusions at many places in the text (e.g., L.293, L.369, 370, 383 and in the labels of Fig. 7. Please check carefully.

(2) Notations will be changed in the revised manuscript as suggested by the reviewer.

*(3)        ($EAC_{OM}$, and $EDC_{OM}$)*

*        $EAC_{OM}$ ($EDC_{OM}$)*

*        $\Delta CO_2 = ((c(CO_2)_{end} * V_{seg}) - (c(CO_2)_{start} * V_{seg})) / \Delta t$* (11)

*        $\Delta CH_4 = ((c(CH_4)_{end} * V_{seg}) - (c(CH_4)_{start} * V_{seg})) / \Delta t$* (12)

*        $\Delta CO_2$ ($\Delta CH_4$)*

*        total C and N*

*        Fig. 7 label: EAC &EDC ($\mu mol\ e^-\ gC^{-1}$)*

(1) 10. L.299. The reference Tamura et al. (1974) only describes the analysis of Fe(II) (albeit in the presence of Fe(III)). How was Fe(III) analyzed?

(2) Fe (III) in the samples was reduced to Fe (II) with 10% ascorbic acid and determined likewise. The procedure will be explained in the revised manuscript as follows:

(3) *"Because 1,10-phenanthroline can only detect $Fe^{2+}$, the $Fe^{3+}$ in the samples was reduced to $Fe^{2+}$ with 10% ascorbic acid. Then, the determined concentration of total Fe was used to calculate the concentration of $Fe^{3+}$ in the samples."*

(1) 11. L.477-479: I cannot follow this sentence. Also compare major point 6 above. Please also note, that Fig. 7 is not mentioned in the text, and that Figure number should be exchanged with that of Fig. 6, since Fig.6 is reported later in the text than Fig. 7.

(2) The order of the figures will be changed and mentioning and numbering of figures will be adjusted in the revised manuscript. The sentence in ll. 447-449 will be changes as follows:

(3) *"$EAC_{OM}$ lay between 218.69 ± 97.15 and 545.71 ± 60.33 $\mu mol\ e^-\ gC^{-1}$ at t0 and decreased on average by 44.85 $\mu mol\ e^-\ gC^{-1}$ until t6. Highest values for $EAC_{OM}$ were found at site 3.125 corresponding to lowest measured $CH_4$ production rates at that site."*

(1) 12.Table 3: Showing the time line as t0, t1, t2 etc. is awkward, since one has to consult the explanation in the methods section. I suggest listing the actual time points, i.e. 0, 1, 3 etc. days.

(2) The captions in Table 3 will be changed as suggested by the reviewer.

(1) 13. Table 4: The numbers in the table show too many decimal positions. Please report only those that are significant. In fact, at numerous places in the text numbers seem to show non-significant decimal positions. Please check and correct.

(2) The numbers in Table 4 as well as other number with non-significant decimal positions will be changed as suggested by the reviewer.

*(3)*

| | CH₄ flux | | | CO₂ flux | | |
|---|---|---|---|---|---|---|
| | rho | p | n | rho | p | n |
| **Clay** | 0.648 | < 0.05 | 12 | 0.605 | < 0.05 | 12 |
| **Silt** | 0.497 | *n.s.* | 12 | 0.302 | *n.s.* | 12 |
| **Sand** | -0.648 | < 0.05 | 12 | -0.605 | < 0.05 | 12 |
| **Fats, waxes, lipids** | -0.833 | < 0.05 | 8 | -0.333 | *n.s.* | 8 |
| **Phenols; humics** | -0.833 | < 0.05 | 8 | -0.357 | *n.s.* | 8 |
| **Aromates** | -0.595 | *n.s.* | 8 | -0.524 | *n.s.* | 8 |
| **Lignin** | -0.786 | < 0.05 | 8 | -0.381 | *n.s.* | 8 |
| **C/N** | -0.881 | < 0.01 | 8 | -0.333 | *n.s.* | 8 |
| **C (%)** | -0.714 | *n.s.* | 8 | -0.190 | *n.s.* | 8 |
| **CH₄ sediment stock change** | -0.222 | *n.s.* | 41 | 0.05 | *n.s.* | 35 |
| **CO₂ sediment stock change** | -0.049 | *n.s.* | 41 | -0.064 | *n.s.* | 35 |

(1) 14.L.535, 538: Should be Table 4 rather Table 5.

(2) The numbering will be changed in the revised manuscript.

(1) 15. References. Some of the references use capital letters for the titles.

(2) Capitals will be changed according to the reviewer's suggestion.

**Authors' response to referee's comments (RC2) on the manuscript, "Organic matter and sediment properties determine in-lake variability of sediment CO2 and CH4 production and emissions of a small and shallow lake" by L.S.E. Praetzel et al.**

Dear reviewer,

Thank you very much for your comprehensive comments on our manuscript "Organic matter and sediment properties determine in-lake variability of sediment CO2 and CH4 production and emissions of a small and shallow lake". We appreciate your suggestions both in form and content and believe that they will substantially improve the paper. In the following, we will outline our responses to your comments one by one and thereby hope to clarify open questions and incorporate your suggestions to your satisfaction.

Each answer will be structured as follows:
(1) comments from referee, (2) author's response, (3) author's changes in manuscript.

Kind regards,

Leandra Praetzel & Co-Authors

**General**

(1) The manuscript needs careful line editing to take care of non-idiomatic English. An example is the frequent usage of wrong tenses (e.g. in line 60: "is mainly depending on" rather than "mainly depends on"). The authors may seek help from a native English speaker for this purpose. I have pointed out a few instances below, but these are by no means exhaustive. I must also concede that I am not a native English speaker!

(2) We asked a native speaker for help and feel certain that her corrections will substantially improve the grammatical style of the paper.

**Specific**

(1) Lines 14-15: Change "... to the atmosphere, following recent studies this is particularly the case for small and shallow lakes." to "... to the atmosphere; recent studies have shown that this is particularly the case for small and shallow lakes."

(2) The sentence will be rephrased in the revised manuscript as follows:

*(3) "Inland waters, particularly small and shallow lakes, are significant sources of carbon dioxide (CO2) and methane (CH4) to the atmosphere."*

(1) Line 16: Delete "yet" and "thus".

(2) The terms will be deleted in the revised manuscript.

(1) Lines 21-22: Change "... were significantly negative (p<0.05, rho<-0.6) correlated" to "... exhibited significant negative correlation (p<0.05, rho<-0.6)". Please make similar changes elsewhere.

(2) The sentence will be rephrased according to the reviewer's suggestion. Similar changes will be made elsewhere in the revised manuscript.

(1) Lines 32-34: The last sentence states the obvious. Who has suggested such a "replacement"?

(2) Some studies implicitly equal production and emission rates, e.g.

Grasset et al. 2018: doi: 10.1002/lno.10786

Sollberger et al. 2014: https://doi.org/10.1007/s00027-013-0319-2

We will adjust the statement in the revised manuscript as follows:

*(3) "We highlight that studies on production rates and sediment quality need to be interpreted with care in terms of deducing emission rates and patterns as it this neglects physical sediment properties and production and oxidation processes in the water column."*

(1) Line 52: Change "has been" to "have been" (here majority is plural), and remove "is".

(2) The term will be changed in the revised manuscript and "is" will be deleted.

(1) Line 56: Remove hyphen between "in" and "lake".

(2) The hyphen will be removed in the revised manuscript.

(1) Lines 58-59: Why is it crucial? Your results show that it is not.

(2) The sentences will be rephrased in the revised manuscript as follows:

*(3) "Nevertheless, anoxic sediments are important for whole lake C cycling as the CO2 and CH4 produced there can be released through the water column to the atmosphere. To understand the spatial patterns of CO2 and CH4 emissions, it is therefore of interest to also understand CO2 and CH4 production processes in the sediment as well as their major controls."*

(1) Line 64: Also its origin (e.g. lignin).

(2) This will be added in the revised manuscript as follows:

*(3) "…and therefore its origin and degree of decomposition…"*

(1) Line 74: Remove "being".

(2) The term will be removed in the revised manuscript.

(1) Line 82: As also pointed out by the other referee a more negative deltaG change would make R2 thermodynamically more favourable.

(2) The statement will be corrected in the revised manuscript.

(1) Lines 86-87: Change "are attributed to" to "may arise from".

(2) The phrasing will be changed in the revised manuscript.

(1) Line 90: Change "remain" to "remains".

(2) The term will be corrected in the revised manuscript.

(1) Lines 92-93 and elsewhere: As also pointed out by the other referee please define each abbreviation when you use it the first time and maintain consistency.

(2) All abbreviations will be defined when being used for the first time and only abbreviations will be used afterwards in the revised manuscript.

(1) Line 96: Why "to a small extent"? In such shallow systems wind-driven turbulence could disturb the sediments. Lines 97-98: Add "penetration" after "oxygen" and remove "in our case". What do you mean by "perennial circulation".

(2) The sentence will be rephrased in the revised manuscript as follows:

(3) *"…but might in the upper parts of the sediments be influenced by oxygen penetration from the water column due to a well-mixed water body."*

(1) Line 99: Please use present indefinite tense, not present continuous.

(2) The tense will be corrected in the revised manuscript.

(1) Line 103: "other" sediment properties?

(2) The term "other" will be added in the revised manuscript.

(1) Line 108: Change "is accountable" by "accounts".

(2) The term will be changed in the revised manuscript.

(1) Lines 110-114: Please rephrase this sentence.

(2) The sentence will be rephrased as follows:

(3) Until now, laboratory incubations of lake sediments were mostly conducted with samples from one or few sites within one lake with a focus on comparing different lakes with each other rather than covering a high in-lake variability of production rates. Further, these studies emphasize temperature effects on production rates (Duc et al., 2010; Gudasz et al., 2010; Gudasz et al., 2015; Fuchs et al., 2016). Unlike peat soils, where a broad range of controls on $CO_2$ and $CH_4$ production has been investigated, to our knowledge, controls such as organic matter (OM) quality, the occurrence of alternative electron acceptors (EAs), thermodynamic processes and sediment grain size have not, or only individually, been systematically surveyed in small lakes.

(1) Line 119: Did you actually investigate "connected productions patterns to OM"?

(2) This sentence might be ambiguous. We will rewrite the sentence as follows:

(3) *"…in order to relate observed production patterns to measured OM and sediment characteristics, thermodynamics, and water-atmosphere fluxes."*

(1) Line 121 and elsewhere: I am not sure if these experiments can be termed as "mesocosm". These were incubations of cores in the lab.

(2) To clarify the experimental procedure and the differences between the two laboratory experiments, we changed the descriptions throughout the whole manuscript: *Sediment incubations* to *slurry incubations* and *Sediment mesocosms* to *Intact sediment core incubations*.

(1) Line 125: Change "hypothesize" to "hypothesized".

(2) The term will be corrected in the revised manuscript.

(1) Line 137: Change "blast" to "blasted".

(2) The term will be corrected in the revised manuscript.

(1) Line 138: Change "arose" to "formed".

(2) The term will be corrected in the revised manuscript.

(1) Figure 1 captions: Technically the depth categories are wrong. For example by <150 cm, you imply depths between 125 and 150 cm, but 20 cm is also <150 cm. This should be clarified (e.g. 125 indicates 100cm<depth_125 cm).

(2) The description of lake depths categories will be clarified in the revised manuscript as follows:

(3) *"…numbers 50, 100, 125 and 150 indicate lake depth category (50: <50 cm, 100: 50-100 cm, 125: 100-125 cm, 150: 125-150 cm)."*

(1) Table 1, caption: Analytical procedures do not have to be mentioned here; they should be described in methodology section.

(2) The description of analytical procedures will be deleted. These can be found in the methods section 2.4 of the revised manuscript.

(1) Line 167: Change "at three occasions" to "on three occasions".

(2) The term will be changed in the revised manuscript.

(1) Line 169: Why randomly? It should be selectively based on a reason.

(2) We randomly chose the sites in order to avoid detecting differences between sites, water depths or transects due to the sampling date. E.g. if we took all four samples from one transect on the same sampling date, we could not have been sure that potentially observed differences in production rates were because of different site characteristics or because of the sampling date. The same would be true for water depths, so we decided to perform a random sampling at each date.

(1) Line 172: Add "respectively" at the end of the sentence.

(2) The term will be added in the revised manuscript.

(1) Line 176: Change "added with" to "containing".

(2) The term will be changed in the revised manuscript.

(1) Line 180: Change "stored" to "maintained".

(2) The term will be changed in the revised manuscript.

(1) Line 192: Referring to a comment by the other reviewer, I note that some isotope data are presented in Table 2 (not for sulphur though), although not at all discussed in the text. It is not clear whether the sample was decalcified. Also what was the reproducibility of measurements? In fact the precision of analysis is not given for any parameter.

(2) Isotopic values of C and N will be mentioned in the revised manuscript in the results sections 3.1.1 and 3.1.2.

Sulphur isotope data was not measured an will therefore be deleted from the methods section.

The samples were not decalcified before analyses. But we analyzed samples for carbonate content which confirmed that carbonate contents were very low (< 0.9 mg/g). We therefore assume that carbonates in the samples only have a minor influence on the results of isotopic data.

During every run of samples, multiple working standards were measured to assure reproducibility of measurements. Precision of standards are: < 1% for C, < 0.1 % for N, < 0.05 ‰ for delta13C, < 0.5 ‰ for delta15N. Information on precision will be added to the methods section 2.2.2 of the revised manuscript.

(1) Line 201: Change "Therefore" to "For this purpose".

(2) The term will be changed in the revised manuscript.

(1) Line 253: Something missing in the sentence.

(2) The term will be corrected in the revised manuscript.

(1) Line 267: Change "analyzed for" to "measured".

(2) The term will be corrected in the revised manuscript.

(1) Lines 274, 291: Change "measured" to "analyzed". (Note samples are analyzed, parameters are measured).

(2) The term will be corrected in the revised manuscript.

(1) Line 301: Change "Therefore" to "For this purpose".

(2) The term will be corrected in the revised manuscript.

(1) Line 313: These are lab experiments, NOT mesocosms!

(2) See comment on line 121.

(1) Line 331: Change "conducted" to "made".

(2) Will be changed in the revised manuscript. Please note that the whole sentence should be rephrased in order to the other reviewer's suggestion as follows:

(3) *"For statistical analyses and discussion, only measurements that were made > 50 days after the deployment of the sediment mesocosms in the climate chamber were used. This was done in order to ensure the system had adapted to experimental conditions and had reached a steady state. Steady state conditions were indicated by quasi-constant $CO_2$ and $CH_4$ concentrations in the sediment."*

(1) Line 361: There is no other way to quantify inputs is ebullition?

(2) The only way to directly quantify ebullition is via inverse funnels that could trap the emitted methane bubbles. We tested this method in our sediment cores, but without success so that we decided to only measure total methane fluxes and separate diffusive and ebullitive fluxes mathematically as this has been suggested by Bastviken et al. 2004 and adapted by many others when measuring in-situ methane fluxes with a floating chamber approach.

(1) Line 408: Change "nor" to "or".

(2) The term will be changed in the revised manuscript.

(1) Fig. 2: Figure difficult to digest. I could not follow "Different letters indicate significant differences between these sites." What do letters "a"-"d" mean? Am I missing anything?

(2) For more clarity, the figure will be split into two and production rates in 5-10 cm depth will be displayed in the supporting information.

a-d denote if there are significant differences between sites. Same letters mean no significant differences. The description will be rephrased in the revised manuscript as follows:

(3) *"Identical lowercase letters indicate production rates that were not significantly different (i.e. $p > 0.05$) from each other."*

(1) Line 483: Change "by averagely" to "on a average by".

(2) The term will be changed in the revised manuscript.

(1) Lines 490, 493, 499: See earlier comments on Lines 21-22.

(2) The sentences will be rephrased according to the above-noted suggestion.

(1) Line 508 and elsewhere: I believe sediment ebbulition in inferred from k>o. I am not sure. Was there any bioturbation that could increase the emission?

(2) Ebullition was inferred from piston velocity k > 2 as described in the methods section 2.3 (see lines 359-361 of the submitted manuscript).

We did not observe any bioturbation during the experiment.

(1) Table 4: "n.s." presumably means not significant (p<0.05). Is is mentioned somewhere? Significance also depends on the number of values that are not given.

(2) n.s. means not significant. An explanation will be added to the table's description as follows:

*(3) "n.s. means that correlations were not significant (p > 0.05)."*

We will revise the table including number of values as follows:

| | CH$_4$ flux | | | CO$_2$ flux | | |
|---|---|---|---|---|---|---|
| | rho | p | n | rho | p | n |
| **Clay** | 0.648 | < 0.05 | 12 | 0.605 | < 0.05 | 12 |
| **Silt** | 0.497 | *n.s.* | 12 | 0.302 | *n.s.* | 12 |
| **Sand** | -0.648 | < 0.05 | 12 | -0.605 | < 0.05 | 12 |
| **Fats, waxes, lipids** | -0.833 | < 0.05 | 8 | -0.333 | *n.s.* | 8 |
| **Phenols; humics** | -0.833 | < 0.05 | 8 | -0.357 | *n.s.* | 8 |
| **Aromates** | -0.595 | *n.s.* | 8 | -0.524 | *n.s.* | 8 |
| **Lignin** | -0.786 | < 0.05 | 8 | -0.381 | *n.s.* | 8 |
| **C/N** | -0.881 | < 0.01 | 8 | -0.333 | *n.s.* | 8 |
| **C (%)** | -0.714 | *n.s.* | 8 | -0.190 | *n.s.* | 8 |
| **CH$_4$ sediment stock change** | -0.222 | *n.s.* | 41 | 0.05 | *n.s.* | 35 |
| **CO$_2$ sediment stock change** | -0.049 | *n.s.* | 41 | -0.064 | *n.s.* | 35 |

(2) Numbers of values for other calculated correlations will be given in the Supporting information Tab. S1.

(1) Lines 528: Change "concentration" to "concentrations" and "was" to "were".

(2) The term will be changed in the revised manuscript.

(1) Line 536: Change "were significantly negative correlated" to "showed significant negative correlation"

(2) The term will be changed in the revised manuscript.

(1) Line 542: What do you mean by "narrower"? lower?

*(2) Narrower* will be changed to *lower* in the revised manuscript.

(1) Lines 545-556: Authors have emphasized on C/N ratio. They have observed increase in C/N with depth in the inner part of the lake. C/N ratio may not be a very efficient parameter to characterize organic source owing to rapid remobilization of nitrogen as well as reabsorption of ammonium on particulates. The paragraph 405 "The C content in the samples was between 2.15 and 33.16% with lowest values at site 3.50 and highest at site 1.50. C/N ratios ranged from 10.97 at site 1.150 to 19.06 at site 3.100. Neither C content nor C/N ratio showed significant changes with sediment nor lake depth, but C/N ratio was significantly higher in samples taken close to the shore (50) than in samples from the lake center (150) ($p < 0.01$)." is very confusing. A graph showing distribution of C/N ratio across the horizontal length of the lake would suitable to comprehend the results better.

(2) We will include a figure showing C/N ratios and absorption ratios for fats/polysaccharides to better illustrate our results.

(3)

[Figure]

(1) Line 551-552: I do not believe in shallow depths it matters.

(2) We additionally propose the mechanism of resuspension and focusing of small particles, that could alter the degree of decomposition of OM.

(3) *"…As this process might not be of the same importance in shallow lakes compared to deeper lakes, we additionally suggest that the more decomposed OM in the lake center might have undergone degradation processes during resuspension and focusing of small particles as a result of wind-induced bed-shearing (Mackay et al., 2011)."*

(1) Line 559: Change "buried" to "getting buried"

(2) The term will be changed in the revised manuscript.

(1) Line 571: Change "role for" to "role in"

(2) The term will be changed in the revised manuscript.

(1) Line 578: Change "e.g." to "among other things"

(2) The term will be changed in the revised manuscript.

(1) Line 587: Change "in the following" to "below"

(2) The term will be changed in the revised manuscript.

(1) Line 591: Remove "the" before "in other studies"

(2) The term will be removed in the revised manuscript.

(1) Lines 612-614: Laborious sentence. All you are saying is that such shallow depths do not get thermally stratified in summer.

(2) We will rewrite the sentence as follows:

(3) *"…especially regarding the fact that shallow waters, as against deeper lakes, do not get thermally stratified in summer and therefore shallow sediment warm much faster (Jankowski et al., 2006)."*

(1) Lines 619-620: Change the tense to present indefinite.

(2) The term will be changed in the revised manuscript.

(1) Line 627-630: What do you mean by "wider" and "narrower"? I do not follow this sentence.

(2) "Wider" and "narrower" will be changed to "higher" and "lower". We will restructure the whole discussions part about OM quality and CO2 and CH4 production rates according to the other reviewer's suggestions. To explain what we mean by this statement: C/N ratios can be interpreted in two ways: a) high C/N ratios = low decomposition state, low C/N ratios = high decomposition state; b) but C/N ratio can also be used to differentiate between OM of terrestrial and aquatic origin (see Meyers 1994) whereas high ratios = terrestrial and low ratios = aquatic origin. It is known that OM of a low decomposition state is easier degradable for microorganisms and therefore leads to higher production rates of CO2 and CH4, but on the other hand, aquatic OM is usually easier degradable for aquatic microorganisms and would therefore lead to higher production rates compared to OM of terrestrial origin (see e.g. Grasset et al. 2018). We therefore conclude that, although there exist two contradicting effects (low vs. high decomposition or aquatic vs. terrestrial origin), the fact that OM closer to the shore is in a lower decomposition state (although it is probably of terrestrial origin) fuels CO2 and CH4 production. The paragraph on C/N ratio and production rates will be revised as follows:

*(3) "C/N ratios are frequently used to characterize the degradation state of OM, but we did not find correlations between C/N ratios and CO2 and CH4 production rates in the slurry incubations. Although OM of autochthonous origin was found to fuel higher degradation rates than allochthonous OM (West et al., 2012; Grasset et al., 2018) we found evidence of predominant inputs of allochthonous (terrestrial) material at sites with higher production rates close to the shore (higher C/N ratios), whereas sites with lower production rates in the lake center received mainly autochthonous (aquatic) OM as indicated by lower C/N ratios (Meyers, 1994). On the other hand, high C/N ratios also indicate a lower degradation state and therefore higher degradation potential whereas low C/N ratios are usually typical of highly decomposed OM having a lower CO2 and CH4 production potential (Malmer and Holm, 1984; Kuhry and Vitt, 1996). These two possibilities of interpreting C/N ratios might be the reason for apparently contradicting findings and the missing relationship between C/N ratios and CO2 and CH4 production rates."*

(1) Line 637: Change the tense to present indefinite.

(2) The term will be changed in the revised manuscript.

(1) Line 654: But the acetate concentration increased!

(2) Please not the discussion in section 4.1.4 of this observation in lines 660-665 of the submitted manuscript.

(1) Line 655: Remove "of" before "importance"

(2) The term will be changed in the revised manuscript.

(1) Line 671: OM quality is not quantified so instead of low you should perhaps use poor.

(2) The term will be changed in the revised manuscript.

(1) Line 673: Change "of energy" to "in energy"

(2) The term will be changed in the revised manuscript.

(1) Line 675: Change "... acetate, but rather is fermentation" to " ... acetate. Instead fermentation may be rate limiting"

(2) The term will be changed in the revised manuscript.

(1) Line 677: Bring "Further" before "it".

(2) The term will be changed in the revised manuscript.

(1) Line 678: Change "finding emphasizes" to "supports"

(2) The term will be changed in the revised manuscript.

(1) Line 692: If the relationship was insignificant the trend cannot be "clear".

(2) The term will be removed in the revised manuscript.

(1) Line 693: Not at all clear, and so is the following conclusion. I find this whole paragraph speculative.

(2) We will rewrite the whole paragraph 4.1.5 - also following the other reviewer's suggestions – where we will elaborate the relationships between $CO_2$ and $CH_4$ production and alternative EAs more precisely. Instead of discussing relationships between EAC and $CH_4$ production, we emphasize that measured inorganic and organic EAs can explain 40-80% of measured $CO_2$ production. The missing capacity can probably be explained by solid-phase iron, which we found ranging from 2 to 3 %, but whereof we do not have information on its speciation.

We further emphasize, that missing correlations between EAC and $CH_4$ production are due to our experimental set-up: the one-week pre-incubation might have already depleted a large amount of reducible organic and inorganic EAs so that subsequent changes and therefore correlations were low.

(1) Line 702: Change "something" to "somewhat"

(2) The term will be changed in the revised manuscript.

(1) Line 706: Change "approaches" to "factors"

(2) The term will be changed in the revised manuscript.

(1) Line 730: Authors attempt to correlate ebullition with grain size. They believe that higher sand content leads to lesser ebullition. Which is highly unlikely since ebullition depends on permeability of sediments and not porosity. Sand always has higher permeability than silt and clay although lesser porosity. You need to elaborate your concept with more clarity

(2) When explaining CH4 ebullition with the concept of grain size distribution/porosity, it is not primarily of importance how permeable the material is, but how effective bubbles can actually accumulate in the sediment (so that they can subsequently be released). Lui et al. 2018 found that the dominant pathway of bubble formation is by displacing the surrounding sediment, and that this is easier in soft, silty sediment compared to sandy sediments. This sediment displacement would lead to more macropores and therefore a higher connectivity creating conduits for bubble release.

We will change the paragraph explaining these mechanisms more precisely as follows:

(3) *"We found ebullition supporting significantly to total $CH_4$ fluxes in two of our four intact sediment core incubations, whereas sites with higher shares of sand exhibited less ebullitive fluxes confirming the findings of Liu et al. (2016) and (2018). The authors explain their findings with the dominant pathway of bubble formation in the sediment, which is by displacing surrounding sediment particles. As this mechanism is more efficient in soft silty sediments compared to sandy material, $CH_4$ bubbles likely accumulate more easily in silt, creating a network of macropores and therefore conduits for subsequent bubble release. We further found OM quality partly exhibiting significant negative correlations with $CH_4$ fluxes, but to a lesser extent than with $CH_4$ production. When preparing slurry incubations, the physical sediment structure is destroyed, so that OM quality becomes the major controlling factor for gas production. These findings suggest that grain size distribution is besides OM quality a main driver of spatial $CH_4$ flux patterns in intact sediment core incubations and that only a combination of physical characteristics and sediment OM quality could sufficiently explain $CH_4$ emission patterns from lakes."*

(1) Line 747: Change "experiment" to "results"

(2) The term will be changed in the revised manuscript.

(1) Line 749: Remove "especially"

(2) The term will be changed in the revised manuscript.

(1) Line 753: Change "vulnerable" to "sensitive"

(2) The term will be changed in the revised manuscript.

(1) Line 754: Change "unroll" to "expect"

(2) The term will be changed in the revised manuscript.

(1) Line 755: Change "lower water columns" to "shallow depths"

(2) The term will be changed in the revised manuscript.

(1) Line 761: Change "refer" to "attribute"

(2) The term will be changed in the revised manuscript.

(1) Lines 764-765: Then why do you not find strong relationship between methane production and (EACorg)?

(2) Please see also comment on Line 693. The statement will be discussed in more detail in the revised manuscript.

(1) Line 770: Measuring "production rate" does not neglect water column processes, interpretation of these data alone would.

(2) The sentence will be rephrased in the revised manuscript as follows:

(3) *"Further, measuring production rates only would neglect the importance of the water column as a sink of sediment generated CH4…"*

**List of relevant changes made in the manuscript**

**Overall**
1. Editing of English language and grammar
2. „...were significantly correlated to...“ was changed to „...exhibited significant negative correlation...“
3. rephrasing of „incubation“ and „mesocosms“ to „slurry incubations“ and intact sediment core incubations“
4. correction of non-significant decimal places
5. revision of use of abbreviations

**Introduction**
L. 105 ff. Correction of standard formation energies
L. 144 ff. Revision of the whole paragraph describing knowledge gaps

**Materials and Methods**
Fig. 1 caption: Renaming of lake depth categories
L. 466 ff.: Adding of description of methods for determining lake water parameters

**Results**
L. 501: Adding of Fig. 2, which displays C/N ratio and fats/polysaccharide ratio at different lake depth categories
Fig. 3: Deletion of production rates from 5-10 cm depth for better clarity
L. 567 ff.: Adding of description of correlations of acetate and hydrogen concentrations with OM quality
L. 610 ff.: Adding of calculation of potential $CO_2$-production from prevalent electron acceptors + Figure

**Discussion**
L. 669 ff.: The chapters „Spatial variability of OM“, „Spatial variability and temperature dependency of $CO_2$ and $CH_4$ production“, and „Influence of OM quality on $CO_2$ and $CH_4$ prodcution rates“ have been merged and therefore shortened
L. 871 ff.: The chapter „Methanogenic pathways“ has been shortened and refocussed on correlations between methanogenic pathways and OM quality
L. 919 ff.: The chapter „Alternative Eas“ has been refocussed; the speculative part on $CH_4$ production has been removed and the discussion on $CO_2$ production has been added
L. 1009 ff.: The discussion on correlation between $CH_4$ bubble formation and grain size distribution has been described in more detail

[revised manuscript text omitted]

For statisticaldata analyses and discussion, we only used measurements that were conductedmade >-50 days after the deployment of the intact sediment core incubations in the climate chamber were used. This was done in order to ensure the system had adapted to experimental conditions and had reached a steady state. Steady state conditions wereAll measurements were conducted after 50 days of incubation of the cores in the climate chamber when the system had reached a steady state as indicated by quasi-constant $CO_2$ and $CH_4$ concentrations in the sediment, as identified by repeated monitoring of dissolved gases.

For the determination ofTo determine $CO_2$ fluxes, the cores were closed gas- tight with a stopper and connected to a laser-based, portable greenhouse- gas analyzer (Los Gatos Research, San Jose, USA), which allowed to-for measure-measuring real-time increase of $CO_2$, $CH_4$ and $H_2O$ concentrations in the headspace of the cores with a resolution of 1 Hz. As the headspace was too small for the instrument's flow rate, a gas bag with a volume of 150 mL was interposed between the headspace and the analyzer. The headspace was closed for 10 minutes, and the diffusive $CO_2$ flux was calculated by linear regression ($R^2 > 0.8$) of using the increase in concentration over time and by the ideal gas law, corrected for air pressure and temperature and related to the water surface area:

$$F = \Delta c / \Delta t * (p * V) / (A * R * T) \tag{8}$$

where F is the $CO_2$ flux in µmol m$^{-2}$ d$^{-1}$, $\Delta c/\Delta t$ is the slope of the linear regression in ppm d$^{-1}$, p is the air pressure in atm, V is the sum of headspace and gas bag volume in m³, A is the water surface area in m², R is the ideal gas constant $8.2*10^{-5}$ m³ atm mol$^{-1}$ K$^{-1}$, and T is the temperature in K.

$CH_4$ fluxes were determined by closing the cores with the stopper for 24 hours, and then taking a gas sample right after closing, and then again after 24 hours with a syringe from the headspace. $CH_4$ fluxes were calculated according to Bastviken et al. (2004):

$$F(CH_4) = k * (C_w - C_{fc}) \tag{9}$$

where F is the $CH_4$ flux in mmol m$^{-2}$ d$^{-1}$, k is the piston velocity in m d$^{-1}$, $C_w$ is the measured $CH_4$ concentration in the water phase in mmol m$^{-3}$ and $C_{fc}$ is the $CH_4$ equilibrium concentration in the headspace at the given $CH_4$ water concentration.

The piston velocity k was determined as:

$$k = (-\ln((c_{sat} - c_{end})/(c_{sat} - c_{start}))/ \Delta t * V) / (A * K_h * R * T) \tag{10}$$

where $c_{sat}$ is the saturation concentration in the chamber headspace at the measured $CH_4$ water concentration, $c_{end}$ is the measured $CH_4$ concentration in the chamber headspace at the end of the flux measurement, and $c_{start}$ is the measured $CH_4$ concentration in the chamber headspace at the beginning of the flux measurement (all in µatm).

The flux was corrected for the non-linear increase of $CH_4$ concentration in the headspace over time due to saturation and divided into diffusive and ebullitive proportions based on the piston velocity (k < 2 = diffusion, k > 2 = ebullition).

**2.3.2 Sediment gas stock change**

$CH_4$ and $CO_2$ concentrations in the sediment were obtained from gas-permeable silicon tubes, determined by gas chromatography as described above (2.2.3) and calculated by Henry's law using  temperature-corrected Henry's constants (see  equation 4). Measured $CO_2$ concentrations were corrected for pH ( equation 5).

The storage change of $CO_2$ and $CH_4$ in the sediment was calculated for each depth segment between two sampling ports as the difference  of $CO_2$ and $CH_4$ concentrations obtained from silicon gas samples at the beginning and at the end of gas flux measurements:

$$\Delta CO_2 \text{\sout{/CH$_4$}} = ((c(CO_2\text{\sout{/CH$_4$}})_{end} * V_{seg}) - (c(CO_2\text{\sout{/CH$_4$}})_{start} * V_{seg})) / \Delta t \tag{11}$$

[revised manuscript text omitted]

Wetzel, R. G.: Gradient-dominated ecosystems: sources and regulatory functions of dissolved organic matter in freshwater ecosystems, in: Salonen K., Kairesalo T., Jones R.I. (eds) Dissolved Organic Matter in Lacustrine Ecosystems, 181–198.

1420    Whiticar, M. J.: Carbon and hydrogen isotope systematics of bacterial formation and oxidation of methane, Chemical Geology, 161, 291–314, https://doi.org/10.1016/S0009-2541(99)00092-3, 1999.

Wik, M., Crill, P. M., Varner, R. K., and Bastviken, D.: Multiyear measurements of ebullitive methane flux from three subarctic lakes, J. Geophys. Res. Biogeosci., 118, 1307–1321, https://doi.org/10.1002/jgrg.20103, 2013.

1425    Wilkinson, J., Maeck, A., Alshboul, Z., and Lorke, A.: Continuous sSeasonal rRiver eEbullition mMeasurements lLinked to sSediment mMethane fFormation, Environ. Sci. Technol., 49, 13121–13129, https://doi.org/10.1021/acs.est.5b01525, 2015.

Yao, H., Conrad, R., Wassmann, R., and Neue, H. U.: Effect of soil characteristics on sequential reduction and methane production in sixteen rice paddy soils from China, the Philippines, and Italy,

1430    Biogeochemistry, 47, 269–295, https://doi.org/10.1007/BF00992910, 1999.

**Supporting Information to: Organic matter and sediment properties determine in-lake variability of sediment $CO_2$ and $CH_4$ production and emissions of a small and shallow lake**

Leandra Stephanie Emilia Praetzel, Nora Plenter, Sabrina Schilling, Marcel Schmiedeskamp, Gabriele Broll, Klaus-Holger Knorr

[Figure]

**Figure S1:** *FTIR spectra of samples from transect 3.*

[Figure]

*Figure S2: CO₂ (a) and CH₄ (b) production rates in 5-10 cm sediment depth. n = 3.*

[Figure]

*Figure S3: CO₂ (a) and CH₄ (b) production rates in the upper 5 cm of the sediment vs. lake depth. n=12. Different letters denote significant differences between groups.*

[Figure]

15 ***Figure*** *__S4__: Spatio-temporal variability of EAC_{OM} (a, b, c), EAC_{tot} (d, e, f) and EDC_{OM} (g, h, i) in the incubation experiment at the beginning (a, d, g, n = 3), and the end (b, e, h, n = 3) of the experiment as well as average values for the whole experiment (c, f, i, n = 6).*

[Figure]

*Figure* S5*: Depth profile of CH₄ (a) and CO₂ (b) concentration in the sediment of the mesocosms. Different symbols denote three replicates at each site. Values are means over sampling period ± SD. n = 2-10*

**Table S1:** Spearman's rank correlation coefficients and significance levels for $CO_2$ and $CH_4$ production, $Q_{10}$-values and EAC and EDC with all other  measured parameters.  n.s. means that correlations were not significant ($p > 0.05$). ACM = acetoclastic methanogenesis, HTM = hydrogenotrophic methanogenesis.

| | $CH_4$ production | | $CO_2$ production | | $Q_{10}$ ($CH_4$) | | acetoclastic methanog. | | EDC | | $EAC_{OM}$ | | $CH_4$ flux | | $CO_2$ flux | |
|---|---|---|---|---|---|---|---|---|---|---|---|---|---|---|---|---|
| | *p* | *rho* | *p* | *rho* | *p* | *rho* | *p* | *rho* | *p* | *rho* | *p* | *rho* | *p* | *rho* | *p* | *rho* |
| aromatics | 0.001 | -0.669 | 0.002 | -0.641 | 0.034 | -0.821 | | | | | | | *0.132* | *-0.595* | *0.197* | *-0.524* |
| fats, waxes, lipids | 0.000 | -0.700 | 0.000 | -0.736 | 0.034 | -0.821 | | | 0.001 | 0.565 | | | 0.015 | -0.833 | *0.428* | *-0.333* |
| humic acids | 0.003 | -0.618 | 0.001 | -0.653 | 0.034 | -0.821 | | | | | | | 0.015 | -0.833 | *0.389* | *-0.357* |
| lignin | 0.003 | -0.606 | 0.003 | -0.610 | 0.034 | -0.821 | | | | | | | 0.021 | -0.786 | *0.352* | *-0.381* |
| phenols | 0.003 | -0.606 | 0.001 | -0.667 | 0.034 | -0.821 | | | | | | | 0.015 | -0.833 | *0.389* | *-0.357* |
| C | *0.194* | *-0.287* | *0.327* | *-0.219* | | | | | | | | | *0.058* | *-0.714* | *0.665* | *-0.190* |
| C/N | *0.356* | *-0.206* | *0.378* | *-0.197* | | | 0.013 | 0.447 | | | | | 0.007 | -0.881 | *0.428* | *-0.333* |
| $H_2$ concentration | 0.000 | 0.450 | 0.000 | 0.515 | | | | | | | *0.157* | *-0.285* | | | | |
| acetate concentration | *0.492* | *0.248* | *0.191* | *0.455* | | | | | | | 0.002 | -0.387 | | | | |
| hydrogenotrophic | | | | | | | | | | | 0.031 | 0.426 | | | | |
| acetoclastic | | | | | | | | | | | *0.748* | *0.042* | | | | |
| $EAC_{OM}$ | *0.946* | *0.030* | *0.470* | *-0.261* | | | | | | | | | | | | |
| $EAC_{inorg}$ | *0.204* | *-0.442* | *1.000* | *0.006* | | | | | | | | | | | | |
| $EAC_{tot}$ | *0.191* | *-0.455* | *0.470* | *-0.261* | | | | | | | | | | | | |
| EDC | 0.031 | -0.697 | 0.039 | -0.673 | | | | | | | | | | | | |
| EAC/EDC | *0.247* | *0.406* | *0.407* | *0.297* | | | | | | | | | | | | |
| EEC | *0.166* | *-0.479* | *0.054* | *-0.636* | | | | | | | | | | | | |
| S (%) | 0.019 | -0.577 | *0.087* | *-0.441* | | | | | | | | | | | | |
| clay | | | | | | | | | | | | | *0.023* | *0.648* | *0.037* | *0.605* |
| silt | | | | | | | | | | | | | *0.101* | *0.497* | *0.340* | *0.302* |
| sand | | | | | | | | | | | | | *0.023* | *-0.648* | *0.037* | *-0.605* |
| $CH_4$ stock change | | | | | | | | | | | | | *0.163* | *-0.222* | *0.714* | *-0.064* |
| $CO_2$ stock change | | | | | | | | | | | | | *0.762* | *-0.049* | *0.776* | *0.050* |

| | CH$_4$ production | | | CO$_2$ production | | | Q$_{10}$ (CH$_4$) | | | EAC$_{OM}$ | | | EAC$_{tot}$ | | | EDC | | |
|---|---|---|---|---|---|---|---|---|---|---|---|---|---|---|---|---|---|---|
| | *p* | *rho* | *n* | *p* | *rho* | *n* | *p* | *rho* | *n* | *p* | *rho* | *n* | *p* | *rho* | *n* | *p* | *rho* | *n* |
| **aromatics** | $< 0.001$ | -0.669 | 22 | $< 0.01$ | -0.641 | 22 | $< 0.05$ | -0.821 | 7 | n.s. | 0.030 | 10 | n.s. | 0.042 | 10 | n.s. | 0.515 | 10 |
| **fats, waxes, lipids** | $< 0.001$ | -0.700 | 22 | $< 0.001$ | -0.736 | 22 | $< 0.05$ | -0.821 | 7 | n.s. | 0.321 | 10 | n.s. | 0.333 | 10 | $< 0.05$ | 0.758 | 10 |
| **humic acids** | $< 0.01$ | -0.618 | 22 | $< 0.01$ | -0.653 | 22 | $< 0.05$ | -0.821 | 7 | n.s. | 0.067 | 10 | n.s. | 0.042 | 10 | n.s. | 0.455 | 10 |
| **lignin** | $< 0.01$ | -0.606 | 22 | $< 0.01$ | -0.610 | 22 | $< 0.05$ | -0.821 | 7 | n.s. | 0.031 | 10 | n.s. | -0.006 | 10 | n.s. | 0.275 | 10 |
| **phenols** | $< 0.01$ | -0.606 | 22 | $< 0.001$ | -0.667 | 22 | $< 0.05$ | -0.821 | 7 | n.s. | 0.152 | 10 | n.s. | 0.115 | 10 | n.s. | 0.479 | 10 |
| **C** | n.s. | -0.287 | 22 | n.s. | -0.219 | 22 | n.s. | -0.393 | 7 | n.s. | -0.394 | 10 | n.s. | -0.370 | 10 | n.s. | -0.321 | 10 |
| **C/N** | n.s. | -0.206 | 22 | n.s. | -0.197 | 22 | n.s. | -0.074 | 7 | n.s. | -0.079 | 10 | n.s. | -0.115 | 10 | n.s. | -0.103 | 10 |
| **δ$^{13}$C** | n.s. | 0.134 | 22 | n.s. | 0.091 | 22 | n.s. | -0.321 | 7 | n.s. | 0.309 | 10 | n.s. | 0.224 | 10 | n.s. | 0.164 | 10 |
| **δ$^{15}$N** | n.s. | -0.267 | 22 | n.s. | -0.281 | 22 | n.s. | -0.071 | 7 | n.s. | 0.055 | 10 | n.s. | -0.006 | 10 | n.s. | -0.164 | 10 |
| **H$_2$ conc.** | $< 0.001$ | 0.450 | 22 | $< 0.001$ | 0.515 | 22 | | | | n.s. | -0.285 | 26 | n.s. | -0.335 | 26 | n.s. | -0.139 | 26 |
| **acetate conc.** | n.s. | 0.248 | 10 | n.s. | 0.455 | 10 | | | | $< 0.01$ | -0.387 | 60 | $< 0.001$ | -0.418 | 60 | n.s. | -0.035 | 60 |
| **HTM** | $< 0.01$ | -0.394 | 22 | $< 0.001$ | -0.516 | 22 | | | | $< 0.05$ | 0.426 | 26 | $< 0.05$ | 0.426 | 26 | n.s. | 0.162 | 26 |
| **ACM** | n.s. | 0.491 | 10 | n.s. | 0.297 | 10 | | | | n.s. | 0.042 | 60 | n.s. | 0.042 | 60 | n.s. | -0.208 | 60 |
| **EAC$_{OM}$** | n.s. | 0.030 | 10 | n.s. | -0.261 | 10 | | | | | | | | | | | | |
| **EAC$_{inorg}$** | n.s. | -0.261 | 10 | n.s. | 0.152 | 10 | | | | | | | | | | | | |
| **EAC$_{tot}$** | n.s. | -0.042 | 10 | n.s. | -0.285 | 10 | | | | | | | | | | | | |
| **EDC** | $< 0.05$ | -0.697 | 10 | $< 0.05$ | -0.673 | 10 | | | | | | | | | | | | |
| **EAC/EDC** | n.s. | 0.406 | 10 | n.s. | 0.297 | 10 | | | | | | | | | | | | |
| **EEC** | n.s. | -0.479 | 10 | n.s. | -0.636 | 10 | | | | | | | | | | | | |
| **S** | n.s. | -0.411 | 16 | $< 0.05$ | -0.446 | 16 | | | | | | | | | | | | |
| **P** | n.s. | 0.282 | 22 | n.s. | 0.305 | 22 | | | | | | | | | | | | |
| **Fe** | n.s. | 0.338 | 22 | $< 0.05$ | 0.453 | 22 | | | | | | | | | | | | |
| **Mn** | n.s. | 0.020 | 22 | n.s. | 0.121 | 22 | | | | | | | | | | | | |
| **CO$_2$ conc.** | | | | | | | | | | n.s. | -0.161 | 20 | n.s. | -0.146 | 20 | n.s. | -0.252 | 20 |
| **CH$_4$ conc.** | | | | | | | | | | n.s. | -0.083 | 20 | n.s. | -0.080 | 20 | n.s. | -0.219 | 20 |

*Table S2: Spearman's rank correlation coefficients and significance levels for Gibb's free energy of acetoclastic (ACM) and hydrogenotrophic (HTM) methanogenesis, acetate and $H_2$ concentrations with OM quality parameters. n.s. means that correlations were not significant ($p > 0.05$).*

| | ACM (t6) | | | acetate conc. (t6) | | | HTM | | | $H_2$ conc. | | |
|---|---|---|---|---|---|---|---|---|---|---|---|---|
| | *p* | *rho* | *n* | *p* | *rho* | *n* | *p* | *rho* | *n* | *p* | *rho* | *n* |
| **aromatics** | n.s. | -0.052 | 30 | < 0.001 | -0.585 | 30 | n.s. | 0.136 | 154 | < 0.05 | -0.187 | 174 |
| **fats, waxes, lipids** | n.s. | -0.237 | 30 | < 0.05 | -0.440 | 30 | < 0.01 | 0.255 | 154 | < 0.001 | -0.352 | 174 |
| **humic acids** | n.s. | 0.082 | 30 | < 0.001 | -0.634 | 30 | < 0.05 | 0.183 | 154 | < 0.01 | -0.229 | 174 |
| **lignin** | n.s. | 0.265 | 30 | < 0.01 | -0.533 | 30 | n.s. | 0.140 | 154 | < 0.05 | -0.190 | 174 |
| **phenols** | n.s. | 0.104 | 30 | < 0.001 | -0.668 | 30 | < 0.05 | 0.196 | 154 | < 0.01 | -0.246 | 174 |
| **C** | n.s. | -0.062 | 30 | n.s. | 0.074 | 30 | < 0.05 | 0.174 | 154 | < 0.05 | -0.149 | 174 |
| **C/N** | < 0.05 | 0.447 | 30 | < 0.01 | -0.538 | 30 | n.s. | -0.057 | 154 | n.s. | 0.102 | 174 |